



# Sedimentation rate and organic matter dynamics shape microbiomes across a continental margin

Sabyasachi Bhattacharya[1], Tarunendu Mapder[2,#,§], Svetlana Fernandes[3,§], Chayan Roy[1,§], Jagannath Sarkar[1,§], Moidu Jameela Rameez[1], Subhrangshu Mandal[1], Abhijit Sar[4], Amit Kumar Chakraborty[5], Nibendu Mondal[1], Sumit Chatterjee[1], Bomba Dam[4], Aditya Peketi[3], Ranadhir Chakraborty[6], Aninda Mazumdar[3,*] and Wriddhiman Ghosh[1,*]

**Author address:**

[1]  Department of Microbiology, Bose Institute, P-1/12 CIT Scheme VIIM, Kolkata - 700054, West Bengal, India.

[2]  Department of Chemistry, Bose Institute, 93/1 APC Road, Kolkata - 700009, India.

[3]  Gas Hydrate Research Group, Geological Oceanography, CSIR National Institute of Oceanography, Dona Paula, Goa - 403004, India.

[4]  Department of Botany, Institute of Science, Visva-Bharati, Santiniketan, West Bengal - 731235, India.

[5]  Department of Environmental studies, Institute of Science, Visva-Bharati, Santiniketan, West Bengal - 731235, India.

[6]  Department of Biotechnology, University of North Bengal, Siliguri, West Bengal 734013, India.

**Present address:**        [#]  Division of Clinical Pharmacology, Department of Medicine, Indiana University School of Medicine, Indianapolis, IN 46202, USA.

[§] Equal contributions

**\* Correspondence:**        wriman@jcbose.ac.in / maninda@nio.org

**Key words:**        marine sedimentary microbiome, sedimentation rate, water-column oxygenation, organic matter deposition and degradation, sedimentary carbon-sulfur cycle, sediment-surface methanogens

**Running Title:**        Sediment microbiomes across a continental margin

---

## Abstract

Marine sedimentation rate and bottom-water $O_2$ concentration control the remineralization/sequestration of organic carbon across continental margins; but whether/how they shape microbiome architecture (the ultimate effector of all biogeochemical phenomena), across shelf/slope sediments, is unknown. Here we reveal distinct microbiome structures and functions, amidst comparable pore fluid chemistries, along 300 cm sediment horizons underlying the seasonal (shallow coastal) and perennial (deep sea) oxygen minimum zones (OMZs) of the Arabian Sea, situated across the western-Indian margin (water-depths: 31 m and, 530 and 580 m,



respectively). The sedimentary geomicrobiology was elucidated by analyzing metagenomes, metatranscriptomes, and enrichment cultures, and also sedimentation rates measured by radiocarbon and lead excess ($^{210}Pb_{xs}$); the findings were then evaluated in the light of the other

geochemical data available for the cores investigated. Along the perennial- and seasonal-OMZ sediment cores, microbial communities were dominated by *Gammaproteobacteria* and *Alphaproteobacteria*, and *Euryarchaeota* and *Firmicutes*, respectively. As a perennial-OMZ signature, a cryptic methane production-consumption cycle was found to operate near the sediment-surface (within the sulfate reduction zone); overall diversity, as well as the relative

abundances of simple-fatty-acids-requiring anaerobes (methanogens, anaerobic methane-oxidizers, sulfate-reducers and acetogens), peaked in the topmost sediment-layer and then declined via synchronized fluctuations until the sulfate-methane transition zone was reached. The entire microbiome profile was reverse in the seasonal-OMZ sediment horizon. In the perennial-OMZ sediments organic carbon deposited was higher in concentration and marine components-

rich, so it potentially degraded readily to simple fatty acids; lower sedimentation rate afforded higher $O_2$ exposure time for organic matter degradation despite perennial hypoxia in the bottom-water; thus, the resultant abundance of reduced carbon substrates sustained multiple inter-competing microbial processes in the upper sediment-layers. Remarkably, the whole geomicrobial scenario was opposite in the sediments of the seasonal/shallow-water OMZ. Our findings create a

microbiological baseline for understanding carbon-sulfur cycling across distinct marine depositional settings and water-column oxygenation regimes.

---

**1. Introduction**

Most of the chemical transformations taking place in marine sediments are functions of *in situ* microbial communities that are co-founded in the sediment system along with the organic matter which is delivered to the seafloor from the water-column (D'Hondt et al., 2019). Depositional dynamics and post-depositional fate of organic carbon in a marine territory depends on a host of

ecosystem properties (LaRowe et al., 2020), of which the *in situ* rate of sedimentation, and dissolved $O_2$ (DO) concentration of the water column, are considered to be of prime importance (Canfield, 1994; Burdige, 2007; Middelburg and Levin, 2009; Ruvalcaba Baroni et al., 2020). Flux and composition of the organic matter and microflora deposited, also influence the carbon remineralization/sequestration dynamics of a sediment system (Kristensen et al., 1995; Parkes et



al., 2000; Burdige, 2007; LaRowe et al., 2020). However, little is known about how microbial community architecture, which is the ultimate driver of all biogeochemical processes, changes in the age-depth context of a diagenetically maturing sediment package (Kallmeyer et al., 2012; Orsi et al., 2017). We also do not have any direct idea about how differential bottom-water DO concentration, sedimentation rate, and flux of labile (biochemically reactive) organic matter, as
often encountered along water-depth transects across continental margins (Middelburg, 2019a, 2019b), shape the structures and functions of marine sedimentary microbiomes. In this scenario, the distinct depositional environments of perennial and seasonal oxygen minimum zones (pOMZs and sOMZs) located in the deep and shallow coastal waters across western continental margins respectively (Naqvi et al., 2000, 2006; Ulloa et al., 2012), afford ideal settings for
geomicrobiological explorations aimed at answering these fundamental questions of marine ecology and biogeochemistry.

      In the partially-landlocked territories of the global ocean, pOMZs occur typically as mid-oceanic bands between 200 and 1200 meters below the sea-level (mbsl) (Lam and Kuypers, 2011). The Arabian Sea pOMZ (AS_pOMZ) is the thickest (vertical span: ~1.2 km), and one of the
largest (~$3.2 \times 10^6$ km$^2$ in terms of total area covered), perpetually oxygen-depleted water mass (<20 µM dissolved O$_2$ round the year) within the global ocean (Ulloa et al., 2012; Acharya and Panigrahi, 2016). It is an outcome of high productivity and biological oxygen demand in the euphotic zone, which is compounded by poor ventilation of the water due to land-locking from three sides; high productivity, in turn, is sustained by monsoon current-driven upwelling of water masses
rich in nitrate, followed by convective mixing during winter (Madhupratap et al., 1996).

      sOMZs build up in thermally-stratified, shallow coastal waters when eutrophication enhances organic matter deposition in tandem with microbial growth and depletion of O$_2$ from the water-column (Levin et al., 2009; Middelburg and Levin, 2009); in case hypoxia persists for years and organic matter continues to accumulate, the OMZ expands and the water becomes euxinic (Turner
et al., 2008). The Arabian Sea sOMZ (AS_sOMZ), which develops transiently over the western Indian shelf due to seasonally-changing costal circulation and hydrography, features eutrophication-induced hypoxia during the summer and autumn months. During the south-west monsoon, the ocean current upwells low-oxygenated water over India's west coast, but the same cannot reach the surface near the shore as a warm freshwater layer is formed over ~10-40 m
water-depths owing to intense coastal rainfall and river drainage (Gupta et al., 2016). Hectic chemoorganoheterotrophic activities add to the intense O$_2$ depletion, often within 10 mbsl water-depth. In this way, by the month of August, the bottom-water becomes suboxic, while in September-October, complete denitrification and sulfate reduction is observed in the water-column;





however with the reversal of surface currents in November-December, oxic condition is reestablished (Naqvi et al., 2000, 2006).

As the two OMZs across the western Indian continental margin feature differential water-column oxygenation regimes, drainage and depositional environments, and marine versus terrestrial organic matter inputs (Fernandes et al., 2018, 2020), here we use their sediment systems as models for investigating microbial community dynamics in distinct diagenetic settings and delineating the physicochemical drivers of microbiome structure and functions across continental shelf/slope sediment horizons. The microbiomes and ecologies were revealed via a "metagenomics - slurry culture - metatranscriptomics" approach, focusing on the population dynamics of sulfate-reducing bacteria and archaea, methanogenic archaea, anaerobically methane-oxidizing archaea-bacteria symbionts (ANME), acetogenic bacteria and anaerobically sulfur-oxidizing bacterial chemolithotrophs (ANSOB). Sedimentation rates of the explored territories were measured via radiocarbon and lead excess dating. These findings were then considered in the context of the geochemical information available for the sediment cores (Fernandes et al., 2018, 2020). Comparison of all microbiological and geochemical data showed that pore-fluid chemistry, which is expected to have profound influence on sedimentary microbiota, is largely comparable for the two physiographically and spatiotemporally distinct oxygenation regimes. Instead, the widely contrasting microbiology of the two sediment systems was shaped by their distinct deposition and degradation dynamics in relation to organic matter.

## 2. Materials and methods

### 2.1 Study area, and sampling

In the course of the research cruise SSK42 (*RV* Sindhu Sankalp 42) the following sediment cores, which form the basis of the current study, were retrieved by gravity coring technique from the upper regions of the western Indian continental slope. SSK42/5 and SSK42/6 were collected from sites located at 580 mbsl (GPS coordinates: 16°49.88' N and 71°58.55' E) and 530 mbsl (GPS coordinates: 16°50.03' N and 71°59.50' E) water-depths, within the eastern AS_pOMZ territory, while SSK42/9 was collected from the AS_sOMZ territory, at a water depth of 31 mbsl (GPS coordinates: 16°13.56' N and 73°19.16' E) (Fig. 1A).

The ~300-cm-long and 12-cm-wide cores were sampled onboard SSK42, at an average resolution of 15-30 cm, as described previously (Bhattacharya et al., 2020; Mandal et al., 2020). For every sediment-depth explored for microbiology, multiple sample-replicates designated





individually for a pair of metagenome, one metatranscriptome, and different culture-based analyses were collected. Sample-replicates were consistently stored at −20°C and 4°C until further use, according as they were designated for culture-independent and culture-dependent studies
respectively. In tandem with sampling for microbiology, individual sediment-depths of the three cores were also sampled, treated, and stored for geochemical analyses, as described previously (Fernandes et al., 2018, 2020; Mandal et al., 2020).

## 2.2 Geological age of the samples and sedimentation rate of the sites

For the two pOMZ cores SSK42/5 and SSK42/6, radiocarbon ($^{14}$C) dates of the sediment samples were estimated in this study according to Stuiver and Polach, (1977), and Fairbanks et al. (2005), as described elsewhere (Bhattacharya et al., 2019). For the sOMZ sediment core SSK42/9, $^{210}$Pb activity in the sediment samples has already been measured by Fernandes et al. (2020) using standard procedure described by Krishnaswami and Lal, (1978); overall sedimentation rate has
been calculated for the core based on Pb excess ($^{210}$Pb$_{xs}$) data, with the extrapolated ages along the core-top to core-bottom trajectory determined from the overall invariant sedimentation rate (Fernandes et al., 2020).

## 2.3 Metagenome sequencing, assembly and annotation

Metagenomes were extracted onboard SSK42 from the designated sample-replicates as described previously (Bhattacharya et al., 2020; Mandal et al., 2020). The duplicate metagenomes isolated in this way for each microbial community explored along the three sediment cores were sequenced separately using Ion PGM or Ion Proton (Thermo Fisher Scientific, Waltham, USA), as described elsewhere (Bhattacharya et al., 2020; Mandal et al., 2020). Each sequence file obtained in this way
(Tables S1-S3) was submitted to Sequence Read Archive (SRA) of National Center for Biotechnology Information (NCBI, Bethesda, USA), with distinct Run accession numbers, under the BioProject PRJNA309469.

For each sedimentary community explored, its quality-filtered (Phred score ≥ 20) metagenomic sequence dataset pair was co-assembled by using the softwares Megahit v1.2.x (Li et al., 2015)
and MetaQUAST (Mikheenko et al., 2016) as described earlier (Bhattacharya et al., 2020; Mandal et al., 2020). Genes, or open reading frames, coding for peptides at least 30 amino acids in length, were identified within contigs having minimum 100 bp length, using MetaGeneMark (Zhu et al., 2010). All the individual gene-catalogs obtained in this way were annotated for the putative functions    of    their    constituent    genes    via    EggNOG    v5.0    database    search


(http://eggnog5.embl.de/download/eggnog_5.0/, last access: 14 April 2020) using EggNOG-mapper and HMMER (Huerta-Cepas et al., 2016).

Parallel to the assembly-based annotation of the metagenomes, the two independent datasets of metagenomic sequence generated for every sedimentary community were individually annotated based on taxonomic affiliation of their constituent reads. For this purpose, the datasets

were searched separately against the NCBI non-redundant (*nr*) protein sequence database (last access: 14 April 2020) as well as the four distinct databases of single-copy conserved marker proteins, which were specially curated from CheckM version 0.7.060 (Parks et al., 2015, last access: 22 December 2020), for all sulfate-reducing, methanogenic, acetogenic or sulfur-oxidizing genera with standing in prokaryotic nomenclature (https://www.bacterio.net/, last access: 22

December 2020). These read matching experiments were carried out using BlastX available within the BLAST+ package (ftp://ftp.ncbi.nlm.nih.gov/blast/executables/blast+/LATEST/), with cut-offs set at minimum 60% identity and 15 amino acid alignment alongside e-value ≤ $1e^{-5}$ (these values are sufficiently stringent to ensure reliable genus-level classification of gene sequences having diverse metabolic function and conservation levels; Mandal et al., 2020). Relative abundance of anaerobic

methanotrophs in a community was determined by searching the corresponding metagenome pair against a manually-curated 16S rRNA gene sequence database that encompassed all the six major groups of ANME (Lloyd et al., 2006). ANME groups and their representative 16S rRNA gene sequences included in the reference database included ANME1a (accession numbers AF419624 and AB019758), ANME1b (accession numbers AF354137, AJ578102, AY324375, AY542191,

AY211707 and AF354126), ANME2a (accession numbers AY592809 and AJ578128), ANME2b (accession numbers AF354128 and AF354138), ANME2c (accession numbers AF419638 and AY211686) and ANME3 (accession numbers AY323224 and AF354136). Read matching experiments against this database were carried out using BlastN available within the BLAST+ package, with 50 nucleotides as the minimum length of alignment, 75% sequence similarity as the

minimum level of matching reads, and e-≤ $1e^{-5}$ as the minimum level of confidence for matching sequences. In all the above mentioned experiments involving local alignment search, for every metagenomic read (query sequence) matching with one or more reference sequence of the database, the best hit was taken. Prior to a matching experiment, all reads present within a metagenomic dataset were trimmed using PRINSEQ (Schmieder and Edwards, 2011) in such a

way as to never contain more than five consecutive bases having Phred scores below 15.

With reference to a metagenome from a given sediment community, percentage allocations of reads to individual taxa/groups were considered to be the direct measures of the relative abundances of the taxa/groups within that community (Tringe et al., 2005; Gill et al., 2006; Jones et





al., 2008; Ghosh et al., 2015; Mandal et al., 2020; Roy et al., 2020). For the genera of sulfate-
reducers, methanogens, acetogens or sulfur-oxidizers, percentage of matching reads was
calculated with respect to the total number of metagenomic reads participating in the search
experiment. For the different ANME groups, percentage of matching reads was calculated with
respect to the total number of 16S rRNA gene sequence reads that were present in the
metagenome being searched (total 16S rRNA gene sequence reads in a metagenome, in turn, was
counted by searching against the RDP database using BLASTN with minimum alignment length:
50 nucleotides, minimum identity cut-off: 75%, and maximum e-value cut-off: $1e^{-5}$). Since duplicate
metagenomes were sequenced for each community, at the end of a read classification experiment
for a given sediment community, two separate values were obtained for the relative abundance
(prevalence) for every taxon/group searched and found to be present in the community. Arithmetic
means of the two independent relative abundance values were calculated (this gave the mean
relative abundance of the taxon/group within the community in question) and used for comparisons
between distinct communities along/across the sediment cores.

For a given sediment community, prevalence of a particular metabolic-type was measured by
summing up the mean relative abundances of all such microbial taxa/groups whose every reported
strain/member is known to exhibit the phenotype(s) of that metabolism. Accordingly, prevalence of
sulfate-reducers was determined by summing up the mean relative abundances of the following
genera - *Desulfurococcus*, *Desulfurolobus* (phylum *Crenarchaeota*), and *Archaeoglobus* (phylum
*Euryarchaeota*) of the domain *Archaea*; the genera *Desulfurobacterium* (phylum *Aquificae*);
*Desulfurispira*, *Desulfurispirillum* (phylum *Chrysiogenetes*); *Desulfacinum*, *Desulfobacca*,
*Desulfobaculum*, *Desulfocurvus*, *Desulfoglaeba*, *Desulfomonas*, *Desulfomonile*, *Desulforhabdus*,
*Desulfosoma*, *Desulfovibrio*, *Desulfovirga*, *Desulfurella*, *Desulfuromonas*, *Desulfuromusa* and
*Thermodesulforhabdus* (class *Deltaproteobacteria*); *Desulfitibacter*, *Desulfitispora*,
*Desulfitobacterium*, *Desulfonispora*, *Desulfosporosinus*, *Desulfotomaculum*, *Desulfurispora*,
*Desulfovirgula*, *Dethiobacter*, *Dethiosulfatibacter*, *Thermodesulfobium* (phylum *Firmicutes*);
*Thermodesulfovibrio* (phylum *Nitrospirae*); *Dethiosulfovibrio* (phylum *Synergistetes*);
*Thermodesulfatator*, *Thermodesulfobacterium* (phylum *Thermodesulfobacteria*) of the domain
*Bacteria*; plus all genera belonging to the families *Desulfarculaceae*, *Desulfobacteraceae*,
*Desulfobulbaceae*, *Desulfohalobiaceae*, *Desulfomicrobiaceae* and *Desulfonatronaceae* of
*Deltaproteobacteria* (references for the sulfate-reducing taxa considered are given in
Supplementary Note 1). Prevalence of methanogenic archaea in a community was determined by
summing up the mean relative abundances of all genera belonging to the classes
*Methanobacteria*, *Methanococci*, *Methanomicrobia* and *Methanopyri* (Whitman et al., 2006).





Prevalence of acetogenic bacteria in a community was determined by summing up the mean relative abundances of the genera *Acetitomaculum, Acetoanaerobium, Acetobacterium,*
*Acetohalobium, Acetonema, Moorella*, *Natroniella, Oxobacter, Ruminococcus*, *Sporomusa* and *Syntrophococcus* (Drake et al., 2006). Prevalence of anaerobically sulfur-oxidizing bacteria in a community was determined by summing up the mean relative abundances of *Beggiatoa*, *Sulfuricurvum*, *Sulfurimonas*, *Sulfurovum*, *Thiobacillus*, *Thioploca* and *Thiomargarita* (references for the genera of ANSOB considered are given in Supplementary Note 2). Prevalence of ANME in
a community was determined by summing up the mean relative abundances of ANME1a, ANME1b, ANME2a, ANME2b, ANME2c and ANME3 (Lloyd et al., 2006).

Distributions of the mean relative abundances of individual metabolic-types along a sediment core were fitted to potential mathematical functions using the software OriginPro 9 as described previously (Fernandes et al., 2018). Attempts were made to fit the fluctuations of the mean relative
abundance data to approximate probability density functions. For this purpose $\chi^2$ values were considered as minimization criteria. $\chi^2$ minimization was achieved by following Levenberg Marquardt Algorithm (Marquardt, 1963; Moré, 1978). For optimal fitting of the *ad hoc* probability density functions to the distribution of the data, the functions were independently iterated up to 4000 times with uniformly sampled parameters considering a tolerance level at $10^{-9}$. The goodness
of all function fittings was reflected in the corresponding minimized $\chi^2$ values.

### 2.4 Quantitative estimation of diversity from taxonomically annotated metagenomic datasets

The level of microbial diversity present in a given sedimentary community was quantified by calculating Simpson Dominance Index, Shannon–Wiener Diversity Index and Shannon–Wiener
Evenness Index (Magurran, 2004) from the mean relative abundances of phyla, determined on the basis of the taxonomic annotation of the corresponding metagenomic data (Ghosh et al., 2015; Roy et al., 2020). Notably, all relative abundance data for the phylum *Proteobacteria* were split into those for the constituent classes. Furthermore, only those groups which had ≥ 0.01 % mean relative abundance in at least one sediment community across the tree cores were considered in
these analyses. Simpson Dominance Index (*D*) for a given community was determined using Equation 1, where $n_i/n$ (denoted as $p_i$) gives the proportion at which the $i^{th}$ phylum is represented in the community (this, in turn, was same as the mean relative abundance of the $i^{th}$ phylum in the community), and S stands for the total phylum count of the community. Shannon Diversity Index (*H*) was calculated using Equation 2: here each $p_i$ value was multiplied by its natural logarithm (Ln
$p_i$), then $p_i \times$ Ln $p_i$  was determined for all the phyla present, and finally $p_i \times$ Ln $p_i$ vaues were summed-up across phyla and multiplied by -1. To determine the level of evenness in the





representation of phyla within a community Shannon Equitability Index ($E_H$) was determined from Equation 3. Here, $E_H$ was calculated by dividing the community's $H$ value by $H_{max}$, which in turn is equal to Ln S (as stated above, S stands for the total phylum count of the community).


$$D = \sum_{i=1}^{s} \left(\frac{n_i}{n}\right)^2 = \sum p_i^2 \qquad \text{Equation 1}$$

$$H = -\sum_{i=1}^{s} p_i \, \text{Ln} \, p_i \qquad \text{Equation 2}$$

$$E_H = \frac{H}{H_{max}} = \frac{H}{\text{Ln S}} \qquad \text{Equation 3}$$

**2.5 Metatranscriptome sequencing and analysis**

Metatranscriptomes were extracted from the designated sample-replicates fixed with RNAlater (Ambion Inc., USA) onboard SSK42, using RNA PowerSoil Total RNA Isolation Kit (MoBio, Carlsbad, USA), while 2×150 nucleotide, paired-end sequencing of the metatranscriptomes was done on a HiSeq4000 platform (Illumina Inc., San Diego, USA), as described elsewhere (Bhattacharya et al., 2020; Mandal et al., 2020).

Although before sequencing potential rRNAs were removed from the native metatranscriptomes using Ribo-Zero Gold (Illumina Inc.), all paired-end metatranscriptomic reads generated for each sedimentary community, before their use in downstream analyses, were mapped on to the rRNA gene sequence database SILVA (Quast et al., 2013) using the software Bowtie2 v.2.3.4.3 (Langmead and Salzberg, 2012) to stamp out whatever rRNA sequence were

potentially still there in the dataset. Subsequent to the *in silico* clean up of the metatranscriptomic sequence datasets they were assembled into contigs using the software utility rnaspades.py, available within the SPAdes package 3.13.0 (Nurk et al., 2013), in default mode. Putative genes, or open reading frames, long enough to code for at last 30 amino acids at a stretch, were identified and reported in >100-bp-long contigs by the use of the software Prodigal v2.6.3 (Hyatt et al., 2010).

Gene-catalogs obtained from individual metatranscriptomes were annotated functionally with the help of the software EggNOG-mapper (Huerta-Cepas et al., 2016) and via searches against the EggNOG v5.0 database using the algorithm HMMER.

Furthermore, each rRNA-read-free sequence dataset was mapped separately (using Bowtie2 v.2.3.4.3) on to five individual genomic sequence databases that represented all sulfate-reducers

(Table S4), methanogens (Table S5), ANME (Table S6), acetogens (Table S7) or ANSOB (Table S8) for which genome sequences were available. Of the five manually-curated databases those for sulfate-reducers, anaerobic sulfur-oxidizers (ANSOB), methanogens and acetogens contained one genome per genus affiliated to the metabolic-type in question; furthermore, only such genera were included under a metabolic-type, all cultured strains of which are known in the literature to possess





that metabolic attribute (whenever genome sequence of the type strain was available, the same
        was selected to represent the genus). While these four databases encompassed 10s of genomes
        and 100s Mb of sequence length, the one for ANME contained only two 3.2 and 3.5 Mb genomes
        and a total of 700 Kb fosmid clone sequence due to the paucity of published ANME genomes. In all
        these operations, Bowtie2 was run in sensitive local read alignment mode, allowing (i) 0

mismatches in seed alignments, (ii) 20 nucleotide seed substrings to align during multiseed
        alignments, (iii) 15 consecutive seed extension attempts to "fail" before Bowtie2 moves on using
        the alignments found until then, and also allowing (iv) Bowtie2 to "re-seed" reads with repetitive
        seeds for a maximum of two occasions. Seed interval function f was put as f(x) = 1 + 0.75 * sqrt (x),
        where x denoted the read length (Langmead and Salzberg, 2012).


**2.6 Enrichment of methanogens and estimation of methane in slurry incubations**

        In order to determine the viability of *in situ* methanogen populations, sediment samples from
        individual depths of the SSK42 cores were added (5 % w/v) to a medium specific for the growth of
        marine methanogens (Whitman et al., 2006), and incubated anaerobically. Each liter of this

medium (pH 7) contained 0.14 g $CaCl_2.2H_2O$, 0.34 g KCl, 0.5 g $NH_4Cl$, 2.75 g $MgCl_2.6H_2O$, 3.45 g
        $MgSO_4.7H_2O$, 0.14 g $K_2HPO_4.3H_2O$, 0.01 g $Fe(NH_4)_2(SO_4)_2.6H_2O$, 0.001 g resazurin, 21.97 g
        NaCl, 2 g yeast extract, 0.5 g $Na_2S$, 0.5 g sodium thioglycolate, 10 ml trace element solution; and 2
        g $NaHCO_3$, 4 g HCOONa, 6.8 g $CH_3COONa$, and 0.04 % (v/v) $CH_3OH$ as methanogenic
        substrates. Notably, this medium contains ~10 mM sulfate ($SO_4^{2-}$) in the form of magnesium and

ferric ammonium salts in addition to the methanogenic substrates formate, acetate and methanol;
        furthermore, the 25-28 mM pore-water sulfates native to the samples were also present in the
        slurry cultures. All but two components of the medium were mixed and autoclaved together in
        screw-capped bottles: only methanol and sodium sulfide were added by means of filter sterilization
        after opening the medium-containing bottles inside an H35 Hypoxystation (Don Whitley Scientific,

West Yorkshire, UK) stipulated at an $O_2$ partial pressure of zero, and temperature 15°C humidity
        75%. Inside the anaerobic workstation, the medium was dispensed into individual culture flasks: 1
        g sediment sample was added to 20 ml medium dispensed in 100 ml narrow-mouth and fixed-joint
        Erlenmeyer flask; all such flasks were then capped by sleeve stopper septa and incubated at 15°C
        for 21 days. Sediment slurry cultures that did not produce methane in the first round of enrichment

were subjected to up to four consecutive sub-cultures by transferring 1 mL clear suspension to
        fresh 20 mL medium (followed by a 21 day incubation) in each round of sub-culturing.
        Concentrations of methane in the head-spaces of all the incubation flasks were determined
        according to Mathew et al. (2015) by injecting 20 µL of the head-space gas into a GC2010 gas





chromatograph (Shimadzu Corporation, Kyoto, Japan) equipped with a thermal conductivity
detector (injector temperature: 200°C; detector temperature: 250°C). An HP-PLOT Molesieve
column (Agilent Technologies, Santa Clara, USA) having 30 m length, 0.32 mm diameter and 12
µm film was used together with Helium as the carrier gas to separate the components of a head-
space gas sample. Temperature of the column was set at 40°C with a 5 min holding-time; it was
subsequently increased to 250°C at a rate of 20°C per 10 min holding-time. Peak areas for
different gases in the chromatographs were calibrated for measuring unknown concentrations by
using a synthetic mixture of nitrogen, hydrogen, carbon dioxide and methane in the ratio 1:1:40:58
by volume.

## 3. Results

### 3.1 Geographical and geological context of the sediment horizons explored
The present study was based on the following sediment cores collected, in the course of the
research cruise SSK42 (*RV* Sindhu Sankalp 42), from the upper regions of the western Indian
continental margin. The cores designated as SSK42/5 and SSK42/6 were collected from sites
located at 580 mbsl (GPS coordinates: 16°49.88' N and 71°58.55' E) and 530 mbsl (GPS
coordinates: 16°50.03' N and 71°59.50' E) water-depths, within the eastern AS_pOMZ territory,
while the core named SSK42/9 was collected from the AS_sOMZ territory, at a water depth of 31
mbsl (GPS coordinates: 16°13.56' N and 73°19.16' E) (Fig. 1A).

For the sediment samples investigated along SSK42/5, radiocarbon([14]C)-based geological
age ranged between approximately 1,000 and 12,000 yr BP; for the samples of SSK42/6, it
spanned between 4,000 and 10,600 yr BP (Figs. 1B and 1C). Sedimentation rate in this territory
ranged between 11 and 132 cm ky$^{-1}$, and there appeared no sign of slumping (age reversal) within
the sediment packages. Notably, sedimentation rate in both the cores increased at depths
corresponding to ~6800 yr BP, and was relatively higher in the more recent, upper layers. On the
other hand, based on Pb excess ($^{210}$Pb$_{xs}$) data (Fernandes et al., 2020), overall sedimentation rate
calculated for SSK42/9 was 0.21 cm y$^{-1}$; core-top to core-bottom ages for this sOMZ sediment
horizon, extrapolated based on a grossly invariant sedimentation rate, spanned between 116.2 and
1487 yr BP (Fig. 1D).

### 3.2 Distinct microbiome compositions characterize AS_pOMZ and AS_sOMZ sediments
On the basis of the data obtained from the taxonomic classification of metagenomic reads,
differentially diversified microbial communities encompassing 40 bacterial/archaeal phyla





(individual classes were considered for the phylum *Proteobacteria*) were detected along the AS_pOMZ sediment cores SSK42/5 and SSK42/6 and the AS_sOMZ sediment core SSK42/9 (Fig. 2). Out of the 40 phyla present at different levels of their relative abundance across the three cores,

17 (*Acidobacteria*, *Actinobacteria*, *Alphaproteobacteria*, *Bacteroidetes*, *Betaproteobacteria*, *Chloroflexi*, *Crenarchaeota*, *Cyanobacteria*, *Deltaproteobacteria*, *Euryarchaeota*, *Firmicutes*, *Gammaproteobacteria*, *Planctomycetes*, *Thaumarchaeota*, *Thermotogae*, *Verrucomicrobia* and *Zetaproteobacteria*) were found to have ≥ 0.1% mean relative abundance in at least one of the explored sedimentary communities of each core. Although these phyla were major constituents of

the microbiome in the pOMZ as well as sOMZ sediment horizons, their distribution pattern varied widely in the two distinct sedimentary systems. For instance, *Gammaproteobacteria* exhibited remarkably high relative abundance within the sedimentary communities of SSK42/5 and SSK42/6, but not SSK42/9; in the three cores, *Gammaproteobacteria* accounted for 16.7-58.6%, 32.2-65.8% and 2.8-23.3% metagenomic reads within individual sedimentary communities respectively (Fig. 2).

*Alphaproteobacteria* was also considerably abundant in the communities of the two pOMZ cores, with sharp increases recorded in the 15-60 cmbsf and 45-60 cmbsf zones of SSK42/5 and SSK42/6 respectively (overall, *Alphaproteobacteria* accounted for 3.9-42.6% metagenomic reads within individual sedimentary communities of the two cores). *Acidobacteria*, *Actinobacteria*, *Bacteroidetes*, *Betaproteobacteria*, *Chlorobi*, *Chloroflexi*, *Cyanobacteria*, *Deltaproteobacteria*,

*Euryarchaeota*, *Firmicutes* and *Planctomycetes* were the other phyla having sizeable representation in both SSK42/5 and SSK42/6 (individually, these phyla accounted for 0.08-24.4% metagenomic reads within the different sedimentary communities explored in the two pOMZ cores). On the other hand, all the communities explored along SSK42/9 are dominated by *Euryarchaeota* and *Firmicutes* (these two phyla accounted for 3.0-26.4% and 7.2-18.5% metagenomic reads

within individual communities of SSK42/9 respectively). Other groups having sizeable representation along the sOMZ core include *Acidobacteria*, *Actinobacteria*, *Alphaproteobacteria*, Unclassified *Archaea*, *Bacteroidetes*, *Betaproteobacteria*, *Chloroflexi*, *Crenarchaeota*, *Cyanobacteria*, *Deltaproteobacteria*, *Gammaproteobacteria*, *Planctomycetes*, *Thaumarchaeota*, and *Thermotogae*; individually, these phyla accounted for 0.1-23.3% metagenomic reads within the

different sedimentary communities explored in SSK42/9.

      For all the 17 major bacterial/archaeal phyla that were detected across the pOMZ and sOMZ sediment horizons, relative abundance fluctuated several times down the sediment-depths in all the three cores (Fig. 2). In SSK42/5 and SSK42/6, relative abundances of most of these phyla declined from the sediment-surfaces to the core-bottoms, while few remained unchanged, and still

fewer increased (for instance, *Gammaproteobacteria* and *Zetaproteobacteria* increased with





sediment-depth in both SSK42/5 and SSK42/6, even though the latter decreased sharply below 250 cmbsf in SSK42/6). Corroborative to these trends, most of the phyla, in the pOMZ cores, had numerically high and statistically significant ($P \leq 0.05$) negative Spearman correlation coefficients ($\rho$) with sediment-depth (Table S9). By contrast, in SSK42/9, relative abundances of many of the

17 major phyla increased steadily with sediment-depth; of these, *Chloroflexi*, *Crenarchaeota*, *Euryarchaeota*, *Firmicutes* and *Thermotogae* had numerically high and statistically significant positive $\rho$ values with sediment-depth (Table S9). Furthermore, *Thaumarchaeota*, and *Korarchaeota*, *Unclassified Archaea*, *Aquificae*, *Deinococcus-Thermus*, *Dictyoglomi*, *Elusimicrobia*, *Fusobacteria* and *Synergistetes* that were sizably present only in SSK42/9, increased with

sediment-depth (Fig. 2).

Phylum-level microbial diversity of individual sedimentary communities, indexed based on their taxonomically-annotated metagenomic data (calculations given in Tables S10-S12), varied considerably along all the three cores (Fig. 3). In the pOMZ cores SSK42/5 and SSK42/6, both Shannon–Wiener Diversity Index ($H$) and Shannon–Wiener Evenness Index ($E_H$) decreased by

~27% from the topmost layers to the core-bottoms; corroboratively, both the indices showed numerically high ($\geq 0.8$) and statistically significant ($P \leq 0.05$) negative $\rho$ values with sediment-depth (Table S13). By contrast, in the same trajectory along the sOMZ core SSK42/9, there was a net increase in $H$ and $E_H$. Notably, however, the overall ranges within which the index values varied in SSK42/9 were quite narrow. Spearman correlations of all the three indices with sediment-depth

were also low for this core (Table S13). In all the three cores, fluctuation of Simpson Dominance ($D$) Index with sediment-depth was inverse to that of $H$ or $E_H$ (Fig. 3).

**3.3 Genes for anaerobic metabolisms related to C-S cycling are abundant across the pOMZ and sOMZ cores**

When the assembled metagenomes of each sediment sample investigated along the two AS_pOMZ cores SSK42/5 and SSK42/6 were annotated individually, 24 contig-collections (out of the total 25 generated) were found to encompass diverse homologs of the sulfate reduction genes which code for sulfate adenylyltransferase (*cysN, cysD, sat and met3*), adenylylsulfate reductase (*aprA* and *aprB*) and dissimilatory sulfite reductase (*dsrA* and *dsrB*) (Table S14). Only for 250

cmbsf of SSK42/6, no homolog was detectable for the different sulfate reduction genes, plausibly owing to relatively low metagenomic data throughput for this sample (Table S2). On the other hand, all the 25 metagenome assemblies obtained from these two cores contained diverse homologs for a large number of the genes involved in acetoclastic methanogenesis; 23 of them encompassed homologs of genes involved in hydrogenotrophic methanogenesis, methylotrophic





methanogenesis as well as biosynthesis of co-enzyme M (Table S15) that is required for methyl group transfer during methanogenesis (Thauer, 1998). All the 25 assemblies also contained diverse homologs of the acetogenesis-related genes *cooS*, *acsA* (encoding anaerobic carbon-monoxide dehydrogenase); *acsB* (encoding acetyl-CoA synthase); *cdhE, acsC* and *cdhD, acsD* (encoding acetyl-CoA decarbonylase/synthase); *acsE* (encoding 5-methyltetrahydrofolate

corrinoid/iron sulfur protein methyltransferase); *fdhA* and *fdhB* [encoding formate dehydrogenase (NADP⁺)]; *fhs* (encoding formate-tetrahydrofolate ligase); *folD* [encoding methylenetetrahydrofolate dehydrogenase (NADP⁺) / methenyltetrahydrofolate cyclohydrolase] and [encoding *metF* methylenetetrahydrofolate reductase (NADPH)] (Table S16). Furthermore, all the 25 metagenome assemblies obtained from the two cores were found to contain diverse homologs of the anaerobic

sulfide oxidation genes which code for sulfide:quinone oxidoreductase (*sqr*) and sulfide dehydrogenase (*fccA* and *fccB*) (Table S17).

        When the assembled metagenomes of each sediment sample explored in the AS_sOMZ core SSK42/9 were annotated individually, all 10 contig-collections generated were found to encompass diverse homologs for large numbers of genes for sulfate reduction (Table S18), hydrogenotrophic

methanogenesis, acetoclastic methanogenesis as well as methylotrophic methanogenesis (Table S19) and acetogenesis (Table S20). 9 out of the 10 metagenome assemblies contained genes required for co-enzyme M biosynthesis (Table S19) and anaerobic sulfur oxidation (Table S21). For the 115 cmbsf sediment sample of SSK42/9, no homolog of co-enzyme M biosynthesis and anaerobic sulfur oxidation genes was detectable; this could be due to the low metagenomic data

throughput obtained for this sample (Table S3).

**3.4 Sulfate-reducers, methanogens, ANME and acetogens predominate in the top-layers of pOMZ sediments and fluctuate synchronously along the cores**

Relative abundances of sulfate-reducers, methanogens, acetogens and ANME were found to vary

in sync with each other throughout the AS_pOMZ cores SSK42/5 (Figs. 4A), and SSK42/6 (Figs. 4B), but not the AS_sOMZ core SSK42/9 (Figs. 4C). Comparable core-wise trends were also observed for the total number of functional gene homologs identified for dissimilatory sulfate reduction, methanogenesis (including hydrogenotrophic, methylotrophic and acetoclastic pathways, plus co-enzyme M biosynthesis) and acetogenesis (i.e., reductive acetyl-CoA pathway or Wood-

Ljungdahl pathway), within the metagenome assemblies obtained for the individual sediment-samples of  SSK42/5, SSK42/6 and SSK42/9 (Table S22). Along SSK42/5 and SSK42/6, relative abundances of sulfate-reducers, methanogens, acetogens and ANME eventually decline from the sediment-surfaces to the core-bottoms, albeit via multiple phases of fall and rise. Corroboratively,





in SSK42/5, and SSK42/6, but not SSK42/9, Spearman correlations between sediment-depth and
prevalence of all these metabolic-types (except for acetogens in SSK42/6) are negative,
numerically high, and statistically significant (Table S23). In SSK42/5 and SSK42/6, prevalence of
all the four metabolic-types individually, are at their respective core-wise maxima within 0-8 cmbsf;
from there they decrease exponentially till the first 60-80 cmbsf. In SSK42/5, the upper
exponential-decay zone is followed by a zone of exponential increase in the relative abundances of
all four metabolic-types; then there are discontinuous reductions in their relative abundances, and
finally Gaussian distributions (Figures 4A and 5). Along SSK42/6, the first exponential-decay zone
is followed by three consecutive zones of discontinuous increase and decay in the relative
abundancse of all four metabolic-types; however, within this territory, only one exponential-decay
zone conform to a probability density function (Figures 4B and 5).

490         In contrast to the above trends, over the first 120 cmbsf of SSK42/9, the trend of fluctuation in
the relative abundance of sulfate-reducers is different from that of methanogens, ANME, and
acetogens (Figs. 4C and 5). While the relative abundances of methanogens, ANME and acetogens
exhibit sharp exponential increases along this sediment-depth, prevalence of sulfate-reducers has
two fluctuation features: an initial weak exponential decay zone overlapped by a subsequent zone
of weak exponential increase that brings the relative abundance of sulfate-reducers almost to the
core-top level. Below 120 cmbsf of SSK42/9, relative abundances of all four metabolic-types
plateau. The exponential decay, exponential increase, and Gaussian distribution, zones identified
across the sediment cores were defined by the equations stated below and numbered as 4, 5 and
6 respectively. The parameters used in these equations, namely $y_0$, $A_1$, $t_1$, $w$, $x_c$, were estimated
simultaneously from the data fitting by $\chi^2$ minimization ($\chi^2$ value for each function fitted is given in
the legend of Figure 5). Consistent with these data, Spearman correlation coefficients ($\rho$) for the
pair-wise associations between (i) sulfate-reducers and methanogens, (ii) methanogens and
ANME, and (iii) ANME and sulfate-reducers, were all found to be higher in SSK42/5 and SSK42/6
than SSK42/9; pair-wise associations of acetogens with the other three metabolic-types were all
individually highest in SSK42/6 (Table S24).

$$\left( y = y_0 + A_1 e^{-\frac{x}{t_1}} \right)$$    Equation 4

$$\left( y = y_0 + A_1 e^{\frac{x}{t_1}} \right)$$    Equation 5

$$\left( y = y_0 + \frac{A}{w\sqrt{\pi/2}} e^{-2\frac{(x-x_c)^2}{w^2}} \right)$$    Equation 6

Throughout SSK42/5 and SSK42/6, sulfate-reducers are most abundant of the four
metabolic-types, followed by ANME, methanogens and acetogens (Figures 4A and 4B); in contrast,





for the most part of SSK42/9 (below 50 cmbsf) ANME predominate over methanogens followed by sulfate-reducers and acetogens (Figure 4C). Although relative abundance of acetogens is lower than sulfate-reducers, methanogens or ANME along all the three sediment cores explored, overall prevalence of acetogens is much higher in SSK42/9 than in SSK42/5 or SSK42/6. Even the lowest

relative abundance of acetogens in SSK42/9 (0.35% at 0 cm) is greater than or almost equal to the highest relative abundances of acetogens in SSK42/5 (0.37% at 0 cm) and SSK42/6 (0.28% at 0 cm) respectively.

### 3.5 Population dynamics of anaerobic sulfur-oxidizing bacteria

Considerable prevalence of ANSOB was detected in all the three cores. In the two pOMZ cores SSK42/5 (Fig. 4D) and SSK42/6 (Fig. 4E), their mean relative abundance in the different sedimentary communities ranges between 0.4-4.6% and 0.3-2.5% of the metagenomic reads annotated respectively; minimum prevalence is encountered within 0-2 cmbsf, while prevalence increases exponentially till 140 and 220 cmbsf in SSK42/5 and SSK42/6 respectively (Fig. 5). In

SSK42/5, the upper zone of exponential increase is followed by a zone of discontinuous reduction in ANSOB-prevalence, and then a Gaussian distribution; in SSK42/6, however, the upper zone of exponential increase is followed by a single zone of sharp exponential decay. On the other hand, overall prevalence, and population distribution (involving a single sharp exponential decay zone), of ANSOB along the sOMZ core SSK42/9 (Fig. 4F) are distinct from SSK42/5 or SSK42/6 (Figs. 5).

ANSOB constitute only 0.4-0.8 % of the communities explored along SSK42/9, except at 0 cmbsf where their prevalence is 1.8%. Notably, core-wise trends of variation comparable to those depicted for ANSOB prevalence in Fig. 4E-F were observed for the total number of functional gene homologs identified for anaerobic sulfur oxidation, within the metagenome assemblies obtained for the individual sediment-samples of  SSK42/5, SSK42/6 and SSK42/9 (Table S22).

Along some segments of SSK42/5 and SSK42/6, but not SSK42/9, trends of fluctuation in the prevalence of ANSOB are reverse to those of sulfate-reducers. However, these dependencies between ANSOB and sulfate-reducers were not reflected in their Spearman correlations determined for the individual sediment cores taken in their entirety (Table S25). This said, in SSK42/5 and SSK42/6 (but not in SSK42/9), $\rho$ value between ANSOB prevalence and sediment-

depth was found to be positive and numerically high; probability value ($P$) corresponding to the $\rho$ value was < 0.05 for SSK42/5 and slightly above this cut-off for SSK42/6 (Table S25). Furthermore, in SSK42/6, fluctuations in the prevalence of ANSOB (Fig. 4E) showed significantly positive correlation (Table S25) with pore-water sulfide ($\Sigma HS^-$) concentration (Fig. 6B), whereas prevalence



of sulfate-reducers (Fig. 4E) showed significantly negative correlation (Table S25) with pore-water
sulfide concentration (Fig. 6B).

### 3.6 Methanogens of the upper layers of AS_pOMZ, but not AS_sOMZ, cores are active *in situ*

The most remarkable ecological feature shared by SSK42/5 (Fig. 4A) and SSK42/6 (Fig. 4B), but
not SSK42/9 (Fig. 4C), was that methanogens within the two pOMZ cores have their maximum
relative abundance at the topmost sediment-layers where, idiosyncratically, there is no free
methane and the abundances of sulfate and sulfate-reducers are also at their core-wise maxima. In
view of their peculiar population ecology in the pOMZ sediments, potential viability of the
methanogens present in two topmost samples of all the three sediment cores were tested via slurry
culture in marine-methanogen-specific medium. Subsequently, *in situ* metabolic functionality of the
viable methanogens populations was checked by metatranscriptome analysis of the native
sediment samples.

After 21 day incubation in methanogen-specific medium at 15°C, samples from 0 and 15
cmbsf of SSK42/5 produced 2.66 and 4.97 µmol methane d$^{-1}$ g sediment$^{-1}$ respectively, 2 and 15
cmbsf of SSK42/6 produced 2.81 and 7.69 µmol methane d$^{-1}$ g sediment$^{-1}$ respectively, whereas 0
and 19 cmbsf of SSK42/9 produced no methane at all. Subsequently, when similar tests were
carried out with the rest of the samples of all the three sediment cores, only those corresponding to
250, 265, 270 and 275 cmbsf of SSK42/6 produced 5.41, 5.82, 4.37 and 3.95 µmol methane d$^{-1}$ g
sediment$^{-1}$ respectively. Furthermore, to test whether very small numbers of viable methanogen
cells were anyhow there in the sediment samples which did not produce any methane in this first
round of slurry culture, the latter were tested for methane production after consecutive rounds of
sub-culturing (enrichment) in marine-methanogen-specific medium (in each sub-culture, 1 ml clear
suspension of the parent culture was transferred to fresh 20 mL medium and incubated for 21
days). Here, only the following samples produced methane, that too after three consecutive sub-
cultures of their initial slurry: 45 cmbsf of SSK42/5 that produced 0.47 µmol methane d$^{-1}$ mL
culture$^{-1}$, and 30 and 235 cmbsf of SSK42/6 that produced 0.21 and 0.29 µmol methane d$^{-1}$ mL
culture$^{-1}$ respectively; the sub-culture experiments were not prolonged any further.

To corroborate the *in situ* functionality of the upper-sediment-layer methanogens of the
AS_pOMZ cores, metatranscriptomes isolated and sequenced from the 0 cmbsf sample of
SSK42/5 and 2 cmbsf sample of SSK42/6 were analyzed for footprints of active methanogens
(since the results of the slurry culture experiments showed that the upper-sediment-layer
methanogens of SSK42/9 were non-viable, metatranscriptomes were not analyzed for the
corresponding samples). Notably, after the elimination of all rRNA-related reads from the native





sequence datasets via mapping against the SILVA database, the 0 cmbsf sample of SSK42/5 and
the 2 cmbsf sample of SSK42/6 (which initially yielded 23,940,274 and 30,010,937 read-pairs) had

retained 23,711,392 and 29,852,795 read-pairs respectively. These two rRNA-sequence-free
metatranscriptomic datasets were individually and separately searched against the comprehensive
genome databases curated for sulfate-reducers, methanogens, ANME, acetogens and ANSOB
(Tables S4-S8). For the 0 cmbsf sample of SSK42/5, 0.42% and 0.02% read-pairs matched
concordantly with sequences present in the genome databases of methanogens and ANME

respectively; 21.73%, 15.36% and 8.0% matched concordantly with sequences present in the
genome databases of sulfate-reducers, ANSOB and acetogens respectively. For the 2 cmbsf
sample of SSK42/6, 0.28% and 0.01% read-pairs matched concordantly with sequences present in
the genome databases of methanogens and ANME respectively; 18.45%, 13.62% and 6.09%
matched concordantly with sequences present in the databases of sulfate-reducers, ANSOB and

acetogens respectively. Notably, for both the samples, very low percentage of metatranscriptomic
reads matched with sequences in the ANME genome database; this was apparently due to the far
smaller size of the ANME database as compared to the other four.

       Furthermore, when the two rRNA-read-free metatranscriptomic sequence datasets were
individually assembled into contigs and annotatedfor putative functional genes, the resultant gene-

catalogs were found to encompass diverse homologs of the (i) sulfate reduction-related genes
*cysN* (encoding sulfate adenylyltransferase subunit 1), *cysD* (encoding sulfate adenylyltransferase
subunit 2) and *aprA* (encoding adenylylsulfate reductase, subunit A) (Table S26); (i) the
methanogenesis-related genes *ackA* (encoding acetate kinase), *pta* (encoding phosphate
acetyltransferase) and *ACSS*/*acs* (encoding acetyl-CoA synthetase) (Table S27); (iii) the

acetogenesis-related genes *fdhA* [encoding formate dehydrogenase (NADP+) alpha subunit], *fhs*
(encoding formate--tetrahydrofolate ligase), *folD* [encoding methylenetetrahydrofolate
dehydrogenase (NADP+) / methenyltetrahydrofolate cyclohydrolase] and *metF*, *MTHFR* [encoding
methylenetetrahydrofolate reductase (NADPH)] (Table S28); and (iv) the anaerobic sulfide
oxidation-related genes *sqr* (encoding sulfide:quinone oxidoreductase), *fccB* [encoding sulfide

dehydrogenase [flavocytochrome c] flavoprotein chain] and *fccA* ( encoding ctochrome subunit of
sulfide dehydrogenase) (Table S29).




## 4. Discussion

### 4.1 Peculiar population ecology of anaerobic microorganisms as a signature of pOMZ sediments

The present exploration of sedimentary microbiota across the western Indian continental margin revealed diverged microbiome architectures in the seasonal (shallow coastal) and perennial (deep sea) OMZs. In the pOMZ and sOMZ sediment horizons, microbial diversity decreased and increased along the cores (Fig. 3), while communities were essentially dominated by *Gammaproteobacteria* and *Alphaproteobacteria*, and *Euryarchaeota* and *Firmicutes*, respectively

(Fig. 2). As a microbiome signature of the pOMZ sediments, methanogens, anaerobic methane-oxidizers, sulfate-reducers and acetogens had their maximum relative abundances in the upper-layers, while prevalence declined with increasing sediment-depth via multiple phases of synchronized fall and rise (Figs. 4A and 4B) until the sulfate-methane transition zone (SMTZ) was reached, as in sediment-depths ≥ 250 cmbsf of SSK42/6 which contained biogenic methane

(Fernandes et al., 2018). Conversely, in the sOMZ sediment horizon explored, prevalence of sulfate-reducers was at its highest, and methanogens, anaerobic methane-oxidizers and acetogens lowest, in the top-layer. Within 50 cm, methanogens, anaerobic methane-oxidizers and acetogens increased sharply while sulfate-reducers decreased slightly; prevalence of all four metabolic-types steadied thereafter (Fig. 4C). Slurry culture and metatranscriptomics showed that

the methanogens of the upper 0-45 cm of the pOMZ, but not sOMZ, cores were functional *in situ*.

For sulfate-reducers, overall decline of their relative abundance in the sediment-surface to core-bottom trajectory, as encountered in the pOMZ sediment system explored (Figs. 4D and 4E), is consistent with the depth-trends of sulfate concentration in SSK42/5 and SSK42/6 (Fig. 6A) as well as global continental slope sediment horizons (Schlesinger and Bernhardt, 2013). However,

the coexistence and covariance of sulfate-reducers with methanogens, ANME and acetogens is idiosyncratic to common ecological axioms since all these metabolic-types employ the Acetyl-CoA pathway for either acetate (biomass) synthesis or acetate degradation, so their natural populations are expected to compete with each other for the common resource hydrogen (Drake et al., 2006). Notably, tandem methanogenesis and sulfate reduction (whether organoclastic or AOM-

dependent) has also been reported from sediment:water interfaces, and upper-sediment-layers well above the SMTZs, of geographically-diverse, organic-matter-rich marine sediments, including those underlying pOMZ waters (Ferdelman et al., 1997; Treude et al., 2005; Mitterer, 2010; Jørgensen and Parkes, 2010; Maltby et al., 2016, 2018; Chronopoulou et al., 2017). Furthermore, biogeochemical features such as shallow depth of SMTZs, and sulfide-build-up (Fernandes et al.,

2018) and cryptic methane cycling within the sulfate reduction zone and near the sediment-surface



(see below), indicate that the microbiome architecture of SSK42/5 and SSK42/6 could be similar to that of the AS_pOMZ segment off the Makran coast of Pakistan (Fischer et al., 2012; Himmler et al., 2015), even as the geodynamics of the cold methane seep sediments off the Makran Coast are distinct from those of the sediments off the west coast of India.

Whereas free methane was there in many of the global methanogenesis sites located within sulfate reduction zones (Maltby et al., 2016, 2018; Chronopoulou et al., 2017), presence of live methanogens across the upper-sediment-layers of SSK42/5 and SSK42/6 is peculiar as there is no methane *in situ* (Fernandes et al., 2018). Metagenomic data, however, indicated that tandem prevalence of ANME, at relative abundance levels greater than those of the methanogens (Figs.

4A and 4B), could be the reason behind the absence of methane in the upper-sediment-layers of AS_pOMZ. In this context it is further noteworthy that for both 0 cmbsf of SSK42/5 and 2 cmbsf of SSK42/6, proportion of metatranscriptomic reads matching with sequences in the ANME genome-database was far less than what mapped on to the methanogens genome-database; this was apparently due to the very small size of the ANME database as compared to the methanogens

database, against which the reads were searched.

     To explain the coexistence of methanogens and sulfate-reducers, we hypothesize that effective hydrogen-crunch in SSK42/5 and SSK42/6, and especially in the upper-sediment-layers of these cores, may not be as acute as the community architecture suggests it to be. For instance, most of the sulfate-reducing genera predominant in these cores have the ability to respire by

reducing sulfur-species other than sulfate (e.g., dimethyl sulfoxide, elemental sulfur, sulfite and/or thiosulfate; see Tables S30 and S31, and references therein), which have less-positive reduction potential than sulfate (Muyzer and Stams, 2008). 50-60% of the methanogens identified in any community of SSK42/5 and SSK42/6 belonged to the family *Methanosarcinaceae*, all members of which can all utilize methylated-substrates (such as methanol and methylamines) without the

requirement for free hydrogen (see Tables S32 and S33, and references therein). Furthermore, many of the methanogenic genera prevalent across SSK42/5 and SSK42/6, remarkably, have hydrogenotrophic and/or acetoclastic methanogenesis reported for all their members; this indicates that there is sufficient supply of hydrogen in this OMZ sediment horizon for multiple apparently-inter-competing hydrogen-requiring processes to proceed unabated. In this context it is noteworthy

that taxonomically-diverse fermentative and exoelectrogenic bacteria, which are potent sources of hydrogen (besides simple carbon sources such as lactate, acetate, $CO_2$, etc.), are also present throughout SSK42/5 and SSK42/6 (Fernandes et al., 2018). Coexistence of acetogens with sulfate-reducers, methanogens and ANME also supports the abundance of hydrogen *in situ* as acetogenic $CO_2$ reduction is known to operate in anoxic environments only when there is a temporal/spatial





relaxation in the competition for hydrogen (Sugimoto and Wada, 1993; Shannon and White, 1996; Hoehler et al., 1999).

Relative abundance of anaerobic sulfur-oxidizers was much higher across the pOMZ cores (Figs. 4D and 4E) as compared to the sOMZ core (Fig. 4F). This indicated that the sulfate-reducers-/methanogens-/ANME-/acetogens-dominated ecology of the pOMZ sediment system was
also sulfur-oxidizers-complemented. The AS_pOMZ (but not AS_sOMZ) cores exhibited significant positive correlation between ANSOB prevalence and sediment-depth, and also ANSOB prevalence and pore-water sulfide concentration; significant negative correlation was observed between prevalence of sulfate-reducers and pore-water sulfide concentration (Table S25). These dependencies could be reflective of the ANSOB recycling some amounts of *in situ* sulfide to sulfate
all through SSK42/5 and SSK42/6, in the same way as they do in the deeper (165-540 cmbsf) layers of pOMZ sediments, off the Peruvian coast (Holmkvist et al., 2011). Such potential sulfide oxidation processes, however, are unlikely to leave isotopic footprint in the sulfide or sulfate present *in situ*, because sulfur-lithotrophic pathways are known to render very small overall-fractionations in the stable isotope ratios of their substrates/products (Alam et al., 2013).


**4.2 Microbial community dynamics within the shallow SMTZ of SSK42/6**

Of the two AS_pOMZ cores, SSK42/6 has detectable build-up of biogenic methane (at 250 cmbsf and below), and thereby a shallower SMTZ, which is apparently a biogeochemical signature of the sediment horizons underlying the heart of the pOMZ's perpendicular span (Fernandes et al., 2018).
Metagenome analyses for the methane-containing samples within the SMTZ of SSK42/6 (Fig. 4B) showed that the steep increase in methane concentration from 38 µM at 250 cmbsf to 65 µM at 265 cmbsf (Fernandes et al., 2018) coincides with sharp increases in the relative abundance of methanogens, as well as sulfate-reducers, ANME and acetogens (notably, at 265 cmbsf of SSK42/6, ~1 mM sulfate is still present in the pore-water; see Fig. 6A). Subsequently, relative
abundances of all four metabolic-types decline sharply at 275 cmbsf concomitant with methane (Fernandes et al., 2018) and sulfate (Fig. 6A) concentrations reaching 952 µM and 0.28 mM (at 280 cmbsf) respectively. These peculiar trends indicate that at 275 cmbsf, acute depletion of sulfate from the pore-water plausibly limits the operation of potential AOM-driven sulfate-reduction. This may be instrumental in the high accumulation of methane, which, in turn, constrains the *in situ*
microbiota, including the methanogens themselves. Absence of $CO_2$ that could have been regenerated from methane if AOM was there, plausibly limits ecosystem productivity further.





### 4.3 Comparable pore-fluid chemistries of the AS_pOMZ and AS_sOMZ sediments

Pore-fluid chemistry along the pOMZ cores SSK42/5 and SSK42/6 has been reported reported in Fernandes et al. (2018). Along these two cores, sulfate concentration decreases linearly with increasing sediment-depth at a gradient of 0.087 mM cm$^{-1}$ and 0.098 mM cm$^{-1}$ respectively (Fig. 6A). Along SSK42/5, sulfur isotope ratio of sulfate ($\delta^{34}S_{SO4}{}^{2-}$) varies from 23.4 ‰ VCDT (at 1 cmbsf) to 45.9 ‰ VCDT (at 280 cmbsf), whereas along SSK42/6, it varies from 23.4 ‰ VCDT (at 1

cmbsf) to 47.4 ‰ VCDT (at 250 cmbsf), with the maximum (51.0 ‰ VCDT) at 235 cmbsf (Fig. 6D). Along SSK42/5, dissolved sulfide ($\Sigma HS^-$) concentration varies from 62.1 µM (at 1 cmbsf) to 54.5 µM (at 280 cmbsf), reaching a maximum of 427 µM at 105 cmbsf sediment-depth; $\Sigma HS^-$ concentration also increases along SS42/6, and reaches a maximum of 2010 µM at a sediment-depth of 250 cmbsf (Fig. 6B). Along SSK42/6, sulfur isotope ratio of dissolved sulfide ($\delta^{34}S_{\Sigma HS}{}^-$) varies from -21.1

‰ VCDT (at 30 cmbsf) to 32.5 ‰ VCDT (at 295 cmbsf), with the minimum (-27.4 ‰ VCDT) recorded at 75 cmbsf (Fig. 6E); $\delta^{34}S_{\Sigma HS}{}^-$ data are unavailable for SSK42/5. Along both the pOMZ cores, concentrations of ammonium ($NH_4^+$) and dissolved inorganic carbon (DIC) increase steadily with depth (Figs. 6C and 6F). Along SSK42/5, $NH_4^+$ varies from 139.3 µM (at 1 cmbsf) to 1596.5 µM (at 280 cmbsf), whereas in SSK42/6 it varies from 382.5 µM (at 1 cmbsf) to 2214.8 µM (at 295

cmbsf). Along SSK42/5, DIC varies from 3.9 mM (at 1 cmbsf) to 15.0 mM (at 280 cmbsf), whereas along SSK42/6, it varies from 3.8 mM (at 1 cmbsf) to 13.1 mM (at 295 cmbsf) with the maximum (14.1 mM) reached at 265 cmbsf.

Pore-fluid chemistry along the sOMZ core SSK42/9 has been reported in Fernandes et al. (2020). Along this core, sulfate concentration also decreases linearly with increasing sediment-

depth (Fig. 6A); the gradient (0.065 mM cm$^{-1}$), however, is less steep as compared to the pOMZ cores. $\delta^{34}S_{SO4}{}^{2-}$, along SSK42/9, increases from 22.5 ‰ VCDT (at 3 cmbsf) to 66.5 ‰ VCDT (at 297 cmbsf) (Fig. 6D). $\Sigma HS^-$ concentration increases with sediment-depth, reaching the maximum (1196.5 µM) at 207 cmbsf (Fig. 6B); $\delta^{34}S_{\Sigma HS}{}^-$ varies from -11.8 ‰ VCDT (at 39 cmbsf) to 6.4 ‰ VCDT (at 297 cmbsf), with the minimum (-22.7 ‰ VCDT) recorded at 54 cmbsf (Fig. 6E). $NH_4^+$ and

DIC concentrations increase steadily with sediment-depth (Figs. 6C and 6F), with $NH_4^+$ from 177.2 µM (at 3 cmbsf) to 2070.3 µM (at 297 cmbsf) and DIC varying from 2.8 mM (at 3 cmbsf) to 19.1 mM (at 297 cmbsf).

### 4.4 Sedimentation rate and organic matter dynamics as drivers of microbiome architecture

The above data and discussions collectively showed that the microbiome and ecology of AS_pOMZ and AS_sOMZ sediment horizons were distinctive despite their comparable pore-fluid chemistries. Remarkably, however, relative abundance, deposition dynamics, composition, and





post-depositional fate of organic matter (Fernandes et al., 2018, 2020) distinguished the two systems significantly. Bottom-water DO level is known not to impact organic matter

degradation/preservation in marine territories (for example, costal locations having shallow water-depths) where sedimentation rate is greater than ~0.04 cm y$^{-1}$ (Canfield, 1994). Most organic carbon in such settings gets buried and preserved, while only small amounts decompose slowly after burial via anaerobic pathways, as pre-burial $O_2$ exposure time is effectively very low irrespective of what amount of $O_2$ is present in the bottom-water (Hartnett et al., 1998);

concurrently, across the global ocean, regardless of the bottom-water DO concentration, organic carbon burial efficiency varies directly and inversely with sedimentation rate and $O_2$ exposure time respectively (Canfield, 1994; Hartnett et al., 1998; Burdige, 2007; Aller, 2014).

In SSK42/5 and SSK42/6, total organic carbon (TOC) content ranges between 1.2 and 4.6 wt %, and 0.6 and 3.7 wt %, respectively; but in SSK42/9 the range of TOC content is smaller (1.3-2.4

wt %) (Fig. 6G). TOC contents of the top-layers of SSK42/5 and SSK42/6 (water-depths: 580 and 530 mbsl respectively) are approximately double that of SSK42/9 (water-depth: 31 mbsl). This is inconsistent with the general water-depth-dependent trend of organic carbon deposition encountered across continental margins. Generally (outside the OMZs), at higher water-depths, the organic detritus gets more time for degradation during transit from the euphotic zone of primary

production to the sea floor, so across the continental margins, organic matter delivery rate decreases with increasing water-depth (Middelburg, 2019a, 2019b). Furthermore, organic carbon flux across the seabed is generally higher in shallower coastal areas, especially within the euphotic zones (up to ~300 mbsl), because productivity is higher in these water-columns, on top of which microphytobenthos, sponges and bioturbating animals augment sediment-surface productivity and

deposition of fresh organic matter that is unreacted upon, so labile or amenable to biodegradation (Middelburg, 2019a, 2019b). In this context, greater amount of organic carbon influx on the pOMZ sediments is potentially attributable to the lack of macrofaunal activity, and low levels of aerobic microbial catabolism, during the passage of the organic matter through the perennially hypoxic water-columns (Cavan et al., 2017; Jessen et al., 2017).

Comparison of the TOC depth-trends of the two sediment systems indicate that, with increasing diagenetic maturity, the organic carbon delivered to the seabed is degraded more rapidly in the pOMZ territory than in the sOMZ. For instance, considering the first 1500 years (up to ~75 cmbsf) of SSK42/5, ~30 % of the deposited TOC is depleted, as compared to ~16 % depletion achieved over the same geological time along the entire length of SSK42/9. While steady TOC

depletion along SSK42/5 and SSK42/6 (Fig. 6G) reflects the labile character of the organic matter deposited in the pOMZ sediments, the more or less unvarying TOC content along SSK42/9 (Fig.




6G) suggests that the organic matter delivered to the sOMZ seafloor is enriched in components refractory to post-depositional degradation. For the sOMZ system, it seems plausible that the labile components of the organic matter have already been degraded in the water-column and sediment:water interface by virtue of exposure to high DO levels, and therefore copious macrofaunal and aerobic microbial activities, for most part of the year (Fernandes et al., 2020). Concurrent to this supposition, molar ratio of TOC and total nitrogen (TN) in the sediment samples, in conjunction with the $\delta^{13}C_{TOC}$ data (Fig. 6H - 6J), indicated that the organic matter present in the pOMZ and sOMZ sediments are predominated by marine and terrestrial components (as per Tyson, 1995), which in turn are more labile and refractory to remineralization respectively (Kristensen et al., 1995; Burdige, 2007).

The above geochemical considerations highlight that the organic matter deposited in AS_pOMZ sediments is not only higher in quantity but also richer in marine biomass than its sOMZ counterpart. As marine organic matter is effectively hydrolyzed into soluble simple fatty acids irrespective of what amount of dissolved $O_2$ is present in the chemical milieu (Burdige, 1991; Kristensen et al., 1995; Aller et al., 1996; Aller and Blair, 2004; Burdige, 2007), its copious delivery on to the pOMZ seafloor, and plausible ready de-polymerization *in situ*, can be instrumental in sustaining high relative abundance of multiple, simple-fatty-acids-requiring metabolic-types (such as methanogens, sulfate-reducers and acetogens) in the top-sediment-layers of SSK42/5 and SSK42/6 (Figs. 4A and 4B), where overall microbial diversity is also at its peak (Fig. 3). Furthermore, the low sedimentation rate (0.011-0.132 cm $y^{-1}$) of this AS_pOMZ territory (Figs. 1B and 1C) may result in an effectively high $O_2$ exposure time (Burdige, 2007) for the degradation of the deposited organic matter, including whatever refractory component may be there, even as DO remains perennially low (~2 μM at the time of current sampling) in the bottom-water (Fernandes et al., 2018). High $O_2$ exposure time, in turn, may usher other biogeochemical mechanisms and conditions that augment organic carbon breakdown (Burdige, 2007), and in doing so enhance the availability of simple fatty acids for methanogens, sulfate-reducers and acetogens in the upper-sediment-layers of SSK42/5 and SSK42/6 (Figs. 4A and 4B). Expectedly, with increasing diagenetic maturity and ageing of sediments in the deeper layers of the pOMZ cores, the residual organic matter becomes increasingly refractory to degradation and reduced metabolites become scarce. This may be the reason behind the overall decrease of methanogens, sulfate-reducers and acetogens along SSK42/5 (Fig. 4A) and SSK42/6 (Fig. 4B), as well as the loss of viability of methanogens in the deeper layers of these cores [notably, methanogens are likely to lose out to sulfate-reducers with increasing competition for reduced metabolites (Whitman et al., 2006)].





On the other hand, the refractory nature of the organic matter deposited in the sOMZ sediments, and consequent shortage of reduced metabolites in the topmost sediment-layer, seems to be the reason why relative abundance of all simple-fatty-acids-requiring anaerobic metabolic-types except sulfate-reducers is lowest at the top-layer of SSK42/9 (Fig. 4C). Notably, within this core, overall microbial diversity is also lowest in the topmost layer (Fig. 3). Albeit high bottom-water

DO (178 µM at the time of current sampling) prevails in this shallow coastal territory for approximately 2/3$^{rd}$ of a year (Naqvi et al., 2006; Fernandes et al., 2020), high sedimentation rate (0.21 cm y$^{-1}$) of the region (Fig. 1D) potentially leads to an effectively low $O_2$ exposure time (Canfield, 1994; Hartnett et al., 1998) for the terrestrial-components-rich organic matter, most part of which would apparently degrade only in the presence of $O_2$ (Kristensen et al., 1995; Burdige,

2007). In this way, the supply of simple fatty acids for methanogens, sulfate-reducers and acetogens get critically limited in the upper-sediment-layers of SSK42/9. However, sharp increase in the relative abundances of methanogens, ANME and acetogens (alongside a small decline of sulfate-reducers) within a few cmbsf of the SSK42/9, followed by steadying of the prevalence of all four metabolic-types (Fig. 4C), signals that oxidative stress eases immediately below the top-layer

and small amounts of the deposited organic matter depolymerizes slowly (plausibly via anaerobic pathways) with increasing diagenetic maturity of the sediment (Hartnett et al., 1998).

    Summing up, the present exploration revealed wide divergence of sedimentary microbiomes in the distinct depositional environments of a seasonal (shallow coastal) and perennial (deep sea) oxygen minimum zone, across a continental margin. Microbiome divergence of the sOMZ and

pOMZ sediment systems was not reflected in their comparable pore-fluid chemistries; instead, distinct organic matter dynamics in relation to its composition, deposition, and post-depositional fate seemed to shape the ecosystems amidst a circuitous influence of water-column DO concentrations. More investigations of fine-resolution organic biogeochemistry are needed for these ecologically critical marine sediment systems to obtain further insights into overall

microbiome evolution in distinct geological settings across the Earth's continental margins.




### Supplementary data

The supplementary materials related to this article are available in the form of an MS Word file named Supplementary Information and one MS Excel file named Supplementary Dataset.


### Data availability

All nucleotide sequence data have been deposited in the Short Read Archive (SRA) of the National Center for Biotechnology Information (NCBI), MD, USA, under the BioProject accession number PRJNA309469: (i) the whole metagenome shotgun sequence datasets have the Run accession numbers SRR3646127 through SRR3646132, SRR3646144, SRR3646145, SRR3646147, SRR3646148, SRR3646150 through SRR3646153, SRR3646155 through SRR3646158, and SRR3646160 through SRR3646165, SRR3570036, SRR3570038, SRR3577067, SRR3577068, SRR3577070, SRR3577071, SRR3577073, SRR3577076, SRR3577078, SRR3577079, SRR3577081, SRR3577082, SRR3577086, SRR3577087, SRR3577090, SRR3577311, SRR3577337, SRR3577338, SRR3577341, SRR3577343 through SRR3577345, SRR3577347, SRR3577349, SRR3577350 and SRR3577351; (ii) the metatranscriptome sequence datasets have the Run accession numbers SRR7990741 and SRR7983762.

### Acknowledgements

We thank the Director CSIR-National Institute of Oceanography (NIO) for facilitating the geochemical studies and the research cruise SSK42 for acquisition of sediment cores. All the support received from the CSIR-NIO Ship Cell members and the crew members of SSK42 is gratefully acknowledged. SB received a fellowship from Bose Institute. SF and JS received fellowships from Council of Scientific and Industrial Research, Government of India (GoI). CR and MJR got fellowships from University Grants Commission, GoI. SM got fellowship from Department of Science and Technology, GoI. NM got fellowships from Science and Engineering Research Board, GoI (under the grant EMR/2016/002703), and Bose Institute.

### Author contributions

WG conceived the study, designed the experiments, interpreted the results and wrote the paper. SB anchored the whole microbiological work, performed the experiments, and analyzed and curated all processed and unprocessed data. AM led the mission SSK42 and all geochemistry studies therein. AM, RC and BD made intellectual contributions to the paper. TM, CR, JS, MJR, SM, AS, AKC, NM and SC performed microbiological experiments and/or data analysis. SF and AP performed geochemical experiments. All authors read and vetted the manuscript.



**Funding**

The microbiological studies were funded by Bose Institute (via intramural faculty grants) and Earth
System Science Organization, Ministry of Earth Sciences (MoES), GoI via the extramural grant
MoES/36/00IS/Extra/19/2013; the research cruise SSK42 was also funded by MoES (GAP2303).

**Competing interest**

The authors declare no competing interest.

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





**Figure legends**

**Figure 1.** Geographical and geological context of the AS_pOMZ and AS_sOMZ sites explored. (A)
Schematic diagram showing the position of SSK42/5, SSK42/6 and SSK42/9 (indicated by green
color), relative to the other SSK42 cores (indicated by pink color) reported elsewhere (Fernandes
et al. 2018, 2020). Water-depth is plotted to scale along the vertical axis of the diagram, while
distances between the cores represented along the horizontal axis are not in scale. Within the
oxygenated water mass (light turquoise shade) the mid-oceanic pOMZ is indicated by blue shade.
Sediment horizons underlying the pOMZ are indicated by gray shade while those impinged by
oxygenated water masses are indicated by brown shade. (B-D) Age versus depth models and
sedimentation rates along (B) SSK42/5 (based on $^{14}$C dates), (C) SSK42/6 (based on $^{14}$C dates)
and (D) SSK42/9 (based on $^{210}$Pb$_{xs}$ data). Data for SSK42/9 were re-plotted from Fernandes et al.
(2020) while those for SSK42/5 and SSK42/6 are from this study.

**Figure 2.** Heat map where the relative abundances of microbial phyla within individual sediment
communities (estimated as the percentages of metagenomic reads affiliated to the phyla upon
searching the datasets against the NCBI *nr* protein database) are compared along, as well as
across, (A) SSK42/5, (B) SSK42/6 and (C) SSK42/9. For each phylum present in a community,
Log$_{10}$ of its mean relative abundance has been plotted in the z axis of the heat map. Only the
phylum *Proteobacteria* has been split into its constituent classes; following this, only such groups
which had ≥ 0.01 % relative abundance in at least one community across the three cores were
considered for the analysis.

**Figure 3.** Simpson Dominance (*D*), Shannon Diversity (*H*) and Shannon Equitability (*E$_H$*) indices of
the individual sediment communities of SSK42/5, SSK42/6 and SSK42/9, determined on the basis
of relative abundances of phyla, which in turn were estimated as the percentages of metagenomic
reads affiliated to the phyla upon searching the datasets against the NCBI *nr* protein database.
Plots corresponded by Spearman's correlation coefficients (*ρ*) ≥ + 0.8 with *P* < 0.05, between the
diversity index concerned and sediment-depth, have green symbols; plots corresponded by
negative *ρ* values numerically ≥ 0.8 with *P* < 0.05, between the diversity index concerned and
sediment-depth, have red symbols; plots corresponded by positive/negative *ρ* values numerically ≤
0.8 have black symbols, irrespective of whether *P* is < 0.05 (all *ρ* values are given in Table S13).





**Figure 4.** Relative abundances of sulfate-reducers, methanogens, anaerobic methanotrophs, acetogens and anaerobic sulfur-oxidizers along (A and D) SSK42/5, (B and E) SSK42/6 and (C and F) SSK42/9. Variations in the relative abundances of sulfate-reducers, methanogens, anaerobic methanotrophs and acetogens are shown in panels A through C, whereas the variations in the relative abundance of anaerobic sulfur-oxidizers are shown (in comparison with sulfur-

reducers) in panels D through F. Relative abundance values plotted for sulfate-reducers, methanogens, acetogens and sulfur-oxidizers are the percentages of metagenomic reads that matched NCBI *nr* protein database sequences from the genera considered as representing these metabolic-types (similar core-wise trends of relative abundance were observed when the percentages of metagenomic reads matching CheckM-derived marker gene sequences from the

genera considered as representing sulfate-reducers, methanogens, acetogens or sulfur-oxidizers were plotted against sediment-depth). Relative abundance values plotted for anaerobic methanotrophs are the percentages of 16S rRNA-encoding metagenomic reads that matched similar sequences from members of the six major groups of ANME. Relative abundance values for the five metabolic-types are plotted in five differently colored symbols. The theoretical lines in the

same color code as the symbols represent the probability density functions simulated for the distribution of the different metabolic-types: solid and dashed lines represent zones of mathematically defined and undefined distribution respectively.

**Figure 5.** Schematic alignment of the different population distribution zones defined by probability density functions, or identified as discontinuous trends, for the relative abundances of sulfate-reducing bacteria and archaea, i.e. prokaryotes (SRP), methanogenic archaea (MGA), archaeal anaerobic methanotrophs (ANME), acetogenic bacteria (AGB) and anaerobic sulfur-oxidizing bacteria (ANSOB), along SSK42/5, SSK42/6 and SSK42/9. The solid lines in three different colors represent three different zones of functional distribution: exponential decay (magenta), exponential

increase (olive), and Gaussian (orange). The solid lines represent the expanses of the mathematically defined population distribution zones; the colored dotted lines represent spans having no mathematically defined distribution of the relevant populations, but, appear to follow the trends of the solid lines having the corresponding colors. The numbers over the solid lines demarcating the zones of functional distribution refer to their reduced $\chi^2$ values; **1:** 0.0621, **2:**

0.0318, **3:** 57.9278, **4:** 0.0160, **5:** 0.0163, **6:** 0.0001, **7:** 0.1743, **8:** 0.0145, **9:** 0.0310, **10:** 0.0007, **11:** 0.0004, **12:** 28.85384, **13:** 0.2110, **14:** 0.8037, **15:** 0.0014, **16:** 0.0183, **17:** 0.0109, **18:** 0.0106, **19:** 0.0042, **20:** 0.0218, **21:** 0.0002, **22:** 0.0002, **23:** 0.1126, **24:** 0.0442, **25:** 0.5305, **26:** 0.10231, **27:** 0.3702, **28:** 0.2281, **29:** 0.0027, **30:** 0.0009.



**Figure 6.** Key parameters of pore-water and solid-phase chemistry along the AS_pOMZ cores SSK42/5 and SSK42/6, and the AS_sOMZ core SSK42/9, compared using data taken from Fernandes et al. (2018) and Fernandes et al. (2020) respectively. (A) Concentration of sulfate ($SO_4^{2-}$), (B) concentration of sulfide ($\Sigma HS^-$), (C) concentration of ammonium ($NH_4^+$), (D) sulfur isotope ratio of sulfate ($\delta^{34}S_{SO4^{2-}}$), (E) sulfur isotope ratio of dissolved sulfide ($\delta^{34}S_{\Sigma HS^-}$), (F) concentration of dissolved inorganic carbon (DIC), (G) TOC content (in wt %), (H) $(TOC/TN)_{molar}$ ratio, (I) carbon isotope ratio of TOC ($\delta^{13}C_{TOC}$), and (J) $\delta^{13}C_{TOC}$ values plotted against $(TOC/TN)_{molar}$ ratio for each sediment sample explored along the three cores. For all the parameters, except $\delta^{34}S_{\Sigma HS^-}$, data have been plotted for all the three cores; only for $\delta^{34}S_{\Sigma HS^-}$, data are unavailable for the pOMZ core SSK42/5.






**Figure 1**



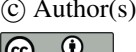



**Figure 2**

Log$_{10}$ of the mean relative abundances of phyla within individual communitites





**Figure 3**

SSK42/5     SSK42/6     SSK42/9

Sediment-depth (cmbsf)



**Figure 4**

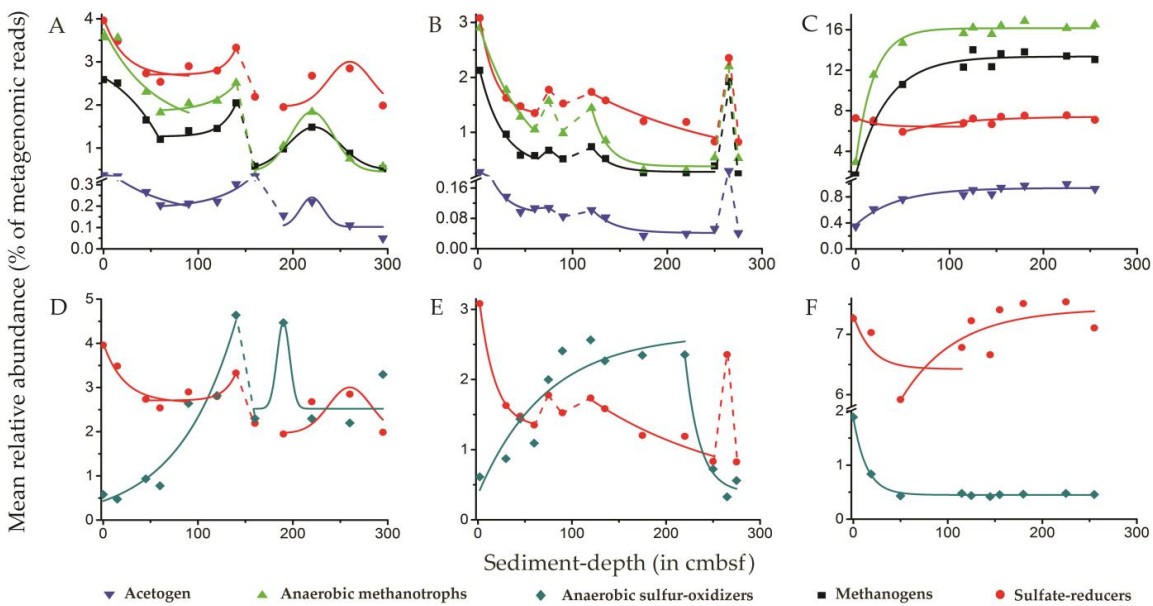

**Figure 5**

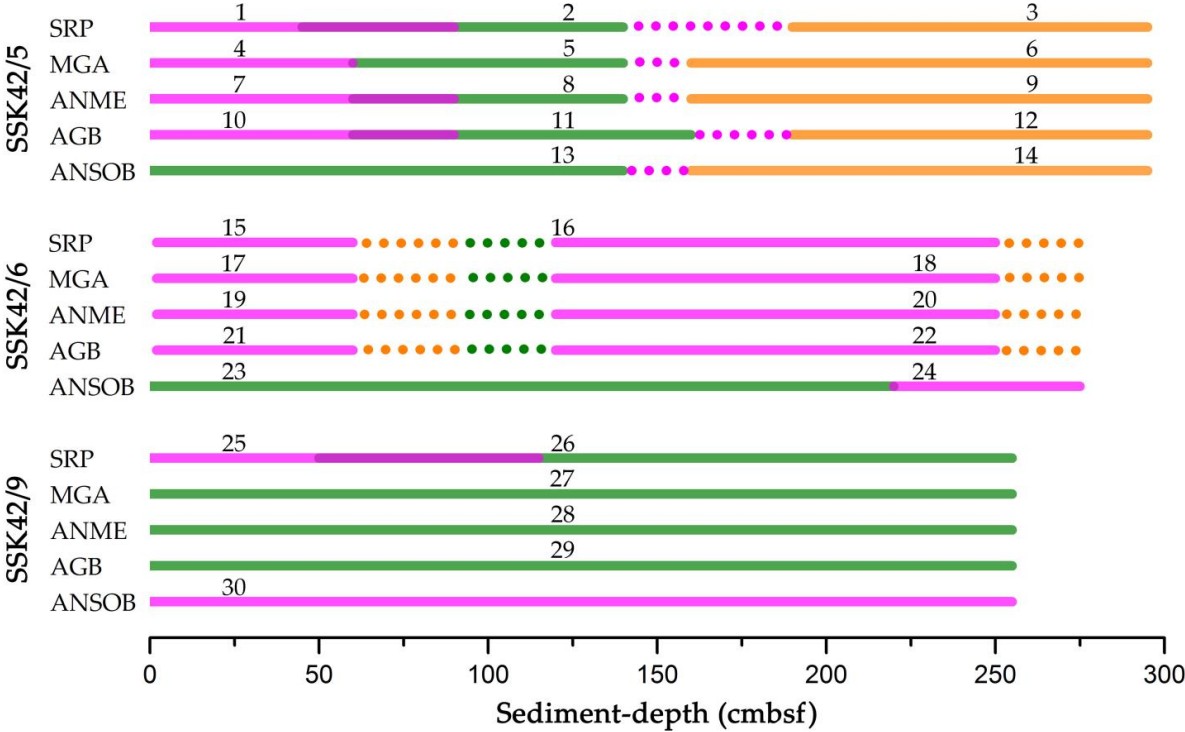



**Figure 6**

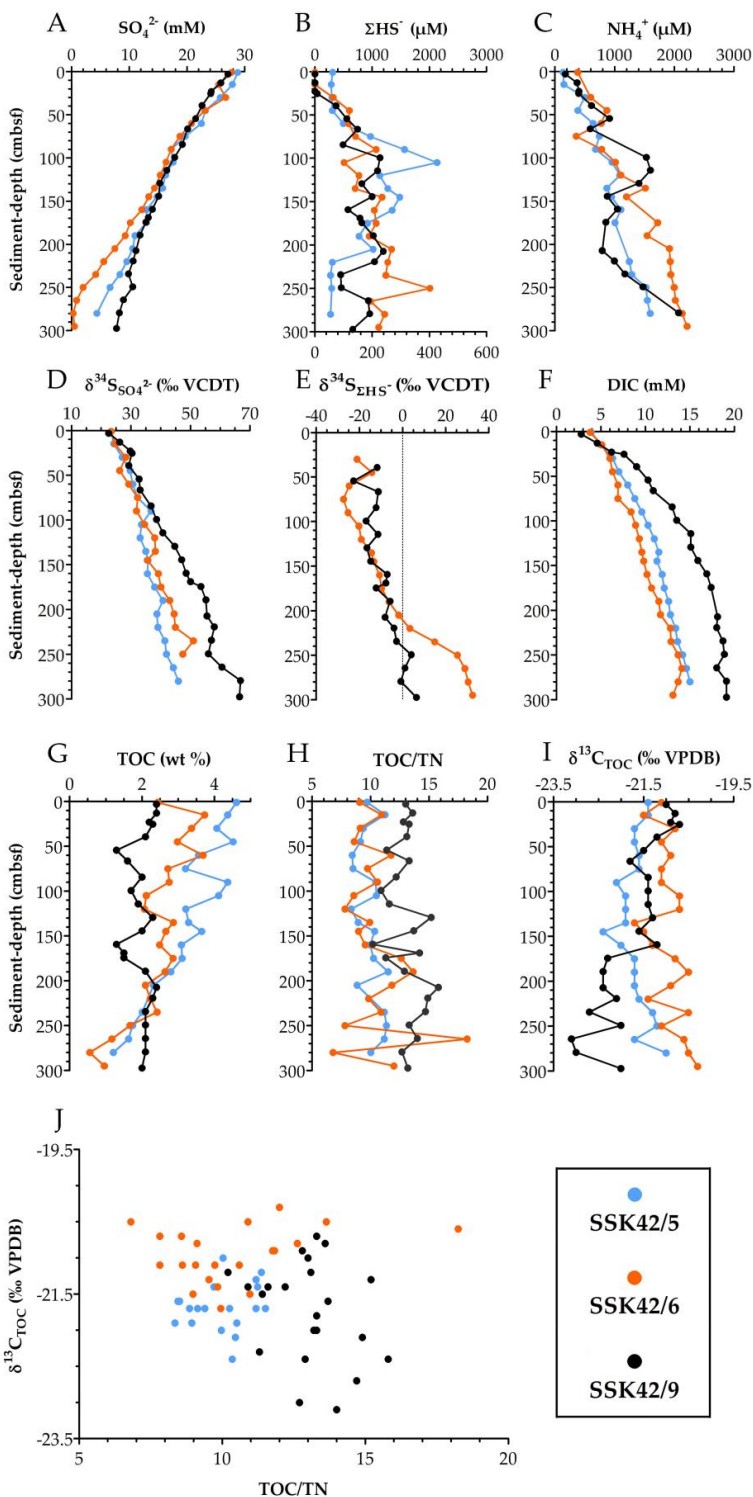