# Peer review of "Sedimentation rate and organic matter dynamics shape microbiomes across a continental margin"

_Biogeosciences, 2021_

## Author Comment (AC1)

**Authors' Response to Anonymous Referee #1**

**General comments**

**Referee's Comment:** Bhattacharya and colleagues obtained a very impressive set of metagenomic and metatranscriptomic data in combination with physicochemical and isotope geochemical measurements. The authors investigate differences in the sediment communities of permanent and seasonal oxygen minimum zones and present shifts in community structure and gene transcripts with depth as well as between the sites. The sampling, the generation, processing and analysis of the data, as well as the sequence archiving was performed carefully. Yet the presentation of the results needs to be improved. In my opinion, the manuscript needs to be shortened (substantially), better structured, and jargon needs to be avoided. The interesting and relevant messages of the study are buried in details, percentages and values. The study is difficult to access in part because the manuscript lacks visualizations (e.g. of the gene transcript abundance). It is recommended to structure all manuscript sections based on compelling objectives, find a red thread that guides the reader through these amazing data, and present only those results that are important to the major findings and conclusions and remove everything else. Please find detailed suggestions below.

**Authors' Response:** We thank the reviewer for appreciating the phenomenology and scientific approaches underlying the study. We also agree with the concerns of the reviewer, so have now dealt with each one of them in the revised manuscript.

**Authors' Changes in Manuscript:** In the Revised Manuscript we have
- re-structured the text by simplifying the language,
- improved the presentation of the results involving percentages and values (by incorporating new figures for better visualization of metatranscriptomic and slurry incubation results),
- moved all the supportive details to the SI, and focused the manuscript towards the main findings of the study.

**Specific comments**

**Referee's Comment:** A picture speaks a thousand words! That is also true for figures. Instead of listing all the taxa, concentrations, abundances etc, please try to visualize. It is so much easier to grasp the critical information that way. For example, the transcript data: The authors have such an incredible dataset. I am wondering why the impressive gene transcript abundance data is not shown in a dedicated figure. The trends become much clearer when all these abundances are shown along the depth profile, or across samples. For inspiration on representation of transcriptomic data check, e.g., these papers:
https://sfamjournals.onlinelibrary.wiley.com/doi/10.1111/1462-2920.14806
https://sfamjournals.onlinelibrary.wiley.com/doi/full/10.1111/1462-2920.13895
https://microbiomejournal.biomedcentral.com/articles/10.1186/s40168-019-0664-z

**Authors' Response:** We agree, and have now edited the manuscript as suggested.

**Authors' Changes in Manuscript:** In the Revised Manuscript we have now
- incorporated new figures for the visualization of the results pertaining to the metatranscriptomic analyses and slurry incubation data (abundances, concentrations, etc.),
- All the lists of taxa that were included to describe the database compositions have now been shifted to the SI,

**Referee's Comment:** L557ff: Can those methane concentrations be shown in a figure?

**Authors' Response:** We agree, now fixed.

**Authors' Changes in Manuscript:** New figures have been included to graphically represent all the concentration values that were involved in the slurry incubation data.

**Referee's Comment:** L583ff: For me, and I think most other readers, it is very difficult to grasp all these percentages and make sense of them, compare them, interpret them. Can this be visualized as well?

**Authors' Response:** We agree, now fixed.

**Authors' Changes in Manuscript:** New figures have been included to graphically represent all the percentage values that were involved in the metatranscriptome analyses.

**Referee's Comment:** Please try to avoid all terms that are very specific for a small area of science (jargon). L479-L489: This section is full of jargon: discontinuous reduction, Gaussian distribution, exponential-decay zone, probability density function, etc. Please avoid using these terms, because most of the readers will not know what they mean. Text sections become very difficult if not impossible to understand when they are full of jargon, and then the entire manuscript becomes hard to digest. In addition, when a reader gets lost in these details the overall message gets lost too. The rule of thumb here is: If for example the Gaussian distribution is really important for the study then it needs to be elaborated and explained with words, concepts and examples everybody understands. If it is not critical for the study, remove it, or move it to the SI for the experts that may be interested in the details.

**Authors' Response:** We agree, so have conceptually elaborated and explained the mathematical terminologies by adding simpler descriptions.
In this context it is noteworthy that the probability density functions (equation fitting) determined and presented for the individual data series critically authenticate the phenomenon "synchronous population fluctuation of multiple metabolic types along the perennial OMZ cores", a main finding of this paper.
If we don't check the population fluctuation trends for potential equations and interpret them mathematically, people would not accept what we are calling increases, decreases or a bell-shaped (Gaussian) distributions are indeed so. From a statistical point of view, people would argue that if still finer resolution data are generated then what is the probability that the trends we are seeing now would still remain so. To answer this it is indispensable to check out probability density functions for the trends observed; if the equations were not obtainable even after carrying out the statistical tests then the envisioned phenomenon of synchronous population fluctuation for the different anaerobic metabolic types along the perennial OMZ sediment cores would have been completely on shaky ground.

**Authors' Changes in Manuscript:** We have now conceptually elaborated and explained the mathematical terminologies by adding simpler descriptions.

**Referee's Comment:** L490-517: This section too is very hard to digest. Is all this detail really necessary? To be honest, I got lost in the equations right away, and was unable to say what the message of the paragraph is even after reading it repeatedly (I couldn't see the proverbial forest from all the trees). Please consider a broad audience.

**Authors' Response:** As stated above, the section between lines 490 and 517 narrates a central finding of the paper: sulfate-reducers, methanogens, ANME and acetogens predominate in the top-layers of perennial OMZ sediments and decline via synchronous fluctuations along the cores. These trends are reverse in the seasonal shallow coastal OMZ sediment horizon, so are crucial for microbiome discrimination in the two sedimentation / oxygenation regimes. Whilst the probability density functions (equation fitting) determined and presented for the individual data series critically authenticate the phenomena, we agree that the statistical concepts introduced needed simpler explanations for general readership.

**Authors' Changes in Manuscript:** We have now conceptually elaborated and explained the mathematical terminologies by adding easier descriptions.

**Authors' Response:** The anaerobic sulfur-oxidizing bacteria not only have crucial roles in the biogeochemistry of these sediment horizons but also hold critical implications for microbiome differentiation in the two sedimentation / oxygenation regimes. Accordingly, it was necessary to explore whether potential statistical correlations existed between their population fluctuations and the population fluctuations of the other metabolic types; also, their population fluctuation trends needed to be authenticated by searching for probability density functions, i.e. finding potential equations that define the trends.

**Authors' Changes in Manuscript:** We have now conceptually elaborated and explained the statistical concepts and mathematical terminologies by adding easier descriptions.

**Referee's Comment:** In addition to an amazing dataset the authors have some very interesting findings. E.g. (L635:"coexistence and covariance of sulfate-reducers with methanogens, ANME and acetogens"). However, those too get lost in all the details. It is highly recommended to shorten and declutter the manuscript. Find out what are the most interesting findings, what is the red thread that keeps the reader's (and writers) attention. Then start by introducing compelling objectives, and then follow these objectives in the results and discussion. Everything that is not needed for the core story should be removed or moved to the SI. There is a lot of potential to shorten the manuscript, I would say it should be at least a third shorter, if not half.

**Authors' Response:** We agree, and have now considerably streamlined the discussion and shortened the manuscript.

**Authors' Changes in Manuscript:** As stated in Authors' Response

**Referee's Comment:** The authors often include results in the discussion section. These sections should either be separated, i.e. the discussion only refers to the results, but contains very little values and numbers anymore. Or they should be combined in a Results and Discussion section. In any case, it would greatly help if the discussion was structured based on objectives that were introduced earlier. Presently the discussion is very broad, and I could not see where the readers attention/focus is drawn towards, i.e. what are the major take home messages.

**Authors' Response:** We agree, and have now considerably streamlined the discussion section.

In this context, however, it is noteworthy that there are no major, or actual, results in the discussion section.

A few current data that were only relevant in the background (perspective) of the discussions (phenomena envisioned) have been provided here in the form of Supplementary Tables.

We also agree that the whole body of geochemical data available for the three sediment cores have been mentioned in the discussion section. But since these are all published data (adequately cited as Fernandes et al., 2018, 2020) they cannot be put under Results; this said, their recapitulation is also central to the current revelation of distinct microbiome structures and functions, amidst comparable pore fluid chemistries, in the sediment horizons underlying the seasonal (shallow coastal) and perennial (deep sea) oxygen minimum zones, so they cannot also be one away with.

**Authors' Changes in Manuscript:** We have now considerably streamlined the discussion section as per the objectives spelt out in the Introduction, and cut back particularly on the detailed values for the geochemical data.

**Referee's Comment:** L615-630: This reads like a summary of the results section. In general, this is welcomed, as now the trends become clear, but it is still a results part, a discussion goes beyond that.

**Authors' Response:** We thank the reviewer for appreciating the importance of having this recapitulative paragraph to spell out the final trends clearly.

**Authors' Changes in Manuscript:** We have further streamlined this paragraph in sync with the objectives of the work and eventual phenomena discovered.

**Referee's Comment:**
L700ff: This contains a lot of results again.
L715-742: Now this section even contains isotope geochemical results that have not yet been mentioned before. There is no mention of a context, so this entire section needs to be moved to the results section. Also, there is way too much information, too many numbers, percentages, etc.
L757ff: Results

**Authors' Response:** As we have already mentioned above, the whole body of geochemical data available for the three sediment cores is given in the discussion section. Since these are all published data (adequately cited as Fernandes et al., 2018, 2020) they cannot be put under Results; yet, their recapitulation is also central to the current revelation of distinct microbiome structures and functions, amidst comparable pore fluid chemistries, in the sediment horizons underlying the seasonal (shallow coastal) and perennial (deep sea) oxygen minimum zones, so they cannot also be removed altogether.

**Authors' Changes in Manuscript:** We have now removed the detailed values for the geochemical data.

**Technical Corrections**

**Referee's Comment:** L38: the influence of bottom water oxygen on benthic microbial communities is not entirely unknown (e.g. Jessen et al. Sci Adv 2017), so it is recommended to write "are less understood" or "are still unclear" etc.

**Authors' Response:** We agree, now fixed.

**Authors' Changes in Manuscript:** As mentioned in the authors' response.

**Referee's Comment:** L81: This section should include mention of the study by Jessen et al 2017 Sci Adv, as it is presenting findings on microbial communities and organic matter in sediments along an oxygen gradient in the Black Sea

**Authors' Response:** We agree, Jessen et al. 2017 now mentioned.

**Authors' Changes in Manuscript:** As mentioned in the authors' response.

**Referee's Comment:** L105: what is meant by hectic? Do you mean substantial?

**Authors' Response:** "Hectic chemoorganoheterotrophic activities" now reworded as "high levels of chemoorganoheterotrophic activities".

**Authors' Changes in Manuscript:** As mentioned in the authors' response.

**Referee's Comment:** L119: ANME only refers to the ANaerobic MEthane oxidizing archaea.

**Authors' Response:** We agree, now fixed.

**Authors' Changes in Manuscript:** "bacteria symbionts" deleted.

**Referee's Comment:** L167: Thank you for depositing the sequence data prior to submission, and also thanks for including all raw data and not just MAGs! That is best practice!

**Authors' Response:** We thank the reviewer for this positive reception.

**Authors' Changes in Manuscript:** Not applicable.

**Referee's Comment:** L227ff: These lists are a good place to start shortening the MS. It could be included in a supplementary table. The idea of summing up the taxa of a target guild may be clear to the reader without the need to present an exhaustive list.

**Authors' Response:** We agree, so have now moved these lists to the SI.

**Authors' Changes in Manuscript:** As mentioned in the authors' response.

**Referee's Comment:** L251: What about ANME-2d (Methanoperedens). It becomes increasingly apparent that this clade may be important at marine systems as well. E.g. in this very recent publication https://www.nature.com/articles/s41396-021-00918-w

**Authors' Response:** We have now upgraded all the metagenomic and metatranscriptomic data pertaining to ANME with analyses based on the searching of our sequence datasets against a comprehensive database of ANME genomes (MAGs) curated non-redundantly from NCBI and GTDB. That curated database contains all the ANME-2d MAGs sequenced thus far.

**Authors' Changes in Manuscript:** As mentioned in the authors' response.

**Referee's Comment:** L270: The equations for the reported Simpson and Shannon indices seem to be the standard equation used for these respective indices. Is there a reason why the authors chose to present the math in detail? Here too it may suffice to cite the respective papers and maybe elaborate on these metrics in the SI. For the reader it would be more helpful to explain the meaning of these metrics, their end members (min/max values) and how to interpret them. In my opinion it is more necessary to explain the terms, .e.g. evenness/equitability than show the equation. As an example, for me it helps to interpret a given index, if I am aware of the max/min values. E.g. that the minimum value Inverse
Simpson can be equals 1 (a pure culture) and the maximum value equals the number of observed species (which is the case when all species occur at exactly the same relative proportion).

**Authors' Response:** We agree, so have now shifted the mathematical details to the SI and provided conceptual explanations and examples in the main text.

**Authors' Changes in Manuscript:** As mentioned in the authors' response.

**Referee's Comment:** L312: From which studies did the two ANME genomes come from? Please reference. There should be quite a number of available ANME MAGs by now. Maybe these can be used as a database for your next studies.

**Authors' Response:** As mentioned above, we have now upgraded all the metagenomic and metatranscriptomic data pertaining to ANME with analyses based on the searching of our sequence datasets against a comprehensive database of ANME genomes (MAGs) curated non-redundantly from NCBI and GTDB. Accordingly, the web address (https://gtdb.ecogenomic.org/) and reference for GTDB (Parks et al., 2018) has been cited in the revised manuscript.

**Authors' Changes in Manuscript:** As mentioned in the authors' response.

**Referee's Comment:** L328: Was hydrogen added as an electron donor for hydrogenotrophic methanogens?

**Authors' Response:** The headspaces of the culture bottles contained hydrogen (10% v/v) as a mixture of $N_2:CO_2:H_2$ = 80:10:10 (v/v/v) was used to deoxygenate the medium, as well as to preset the Whitley H35 Hypoxystation (inside which inoculum addition, culture incubation and culture transfer were carried out) at 0% partial pressure of $O_2$.

**Authors' Changes in Manuscript:** The above details are already there in the reference that was cited for the methanogen-specific medium used (Whitman et al., 2006); still we have now mentioned these points in the revised text over and above the existing reference.

**Referee's Comment:** L356ff: A lot of the results in this section are redundant to the Sampling section in the Materials and Methods. Please shorten, or remove, at either section.

**Authors' Response:** We agree, now fixed.

**Authors' Changes in Manuscript:** Redundant information now removed from the Results section.

**Referee's Comment:** L364: What does BP stand for, please write out each abbreviation at first use.

**Authors' Response:** We agree, now fixed.

**Authors' Changes in Manuscript:** Full form of BP (before present) has now been provided.

**Referee's Comment:** L379: Which database and version was used for the taxonomic classification? I might have missed this in the methods, but if it is not mentioned, please include.

**Authors' Response:** For taxonomic classification of metagenomic reads the datasets were searched separately against the NCBI non-redundant (*nr*) protein sequence database (last access: 14 April 2020) as well as the four distinct databases of single-copy conserved marker proteins, which were specially curated from CheckM version 0.7.060 (Parks et al., 2015, last access: 22 December 2020).

**Authors' Changes in Manuscript:** This was already mentioned in Methods.

**Referee's Comment:** L387: Please refer to "relative sequence abundance" to avoid confusion with other measures of relative abundance.

**Authors' Response:** We agree, now fixed.

**Authors' Changes in Manuscript:** Text edited as suggested.

**Referee's Comment:** L392f: Generally, it is recommended to phrase it the other way around: "…xx % of metagenomic reads affiliated with Alphaproteobacteria…"

**Authors' Response:** We agree, now fixed.

**Authors' Changes in Manuscript:** Text edited as suggested.

**Referee's Comment:** L393ff: Personally, I think it is not necessary to list all these taxa. It makes the manuscript very long and voluminous and disrupts the reading flow. Unless a specific taxon needs to be

highlighted, it will suffice to point the reader to a figure showing the clades and their relative sequence abundances.

**Authors' Response:** We agree and have now reduced the text as suggested.

**Authors' Changes in Manuscript:** Mention of all the relatively-less abundant taxa have been removed from the revised text and the overall difference in the microbiome profiles of perennial and seasonal OMZ sediments has been pointed out by referring to the heat map of Figure 2.

**Referee's Comment:** L480: I think microbial guilds would be a more widespread and better term than "metabolic-types"

**Authors' Response:** We agree that microbial guilds is a widespread terminology but since it does not specify that the groups here were unified based on particular metabolisms we prefer to retain the term "metabolic-types".

**Authors' Changes in Manuscript:** Not applicable.

**Referee's Comment:** L579: The number of reads is important, but not here. 1. Large numbers break the reading flow and 2.) including the read numbers does not add info that is necessary for the argument at this point. This section is about mcrA transcripts and so it should be focused on this.

**Authors' Response:** We agree, and have now moved the information on read numbers to Methods section.

**Authors' Changes in Manuscript:** As mentioned in the authors' response.

**Referee's Comment:** L580ff: redundant to materials section. Please remove.

**Authors' Response:** We agree, now fixed.

**Authors' Changes in Manuscript:** This redundant portion has been removed from revised manuscript.

**Referee's Comment:** L652: Could the absence of methane not be explained also by aerobic methane oxidation?
Did you check if you have aerobic methane oxidizers in the dataset (eg Methylococcales).

**Authors' Response:** We agree and thank the reviewer very much for vindicating our previous findings regarding aerobic methane oxidation in the same sulfidic cores SSK42/5 and SSK42/6 (please see Bhattacharya et al.: Aerobic microbial communities in the sediments of a marine oxygen minimum zone. FEMS Microbiol. Letts., 367, fnaa157, 2020).

**Authors' Changes in Manuscript:** The revised text now factors in the role of aerobic methane oxidizers by citing Bhattacharya et al., 2020.

**Referee's Comment:** L659: Is there a reason why a database like GTDB (https://www.nature.com/articles/nbt.4229?testing) was not used? It should contain plenty of ANME genomes/MAGs.

**Authors' Response:** We have now upgraded all the metagenomic and metatranscriptomic data pertaining to ANME with analyses based on the searching of our sequence datasets against a comprehensive database of ANME genomes (MAGs) curated non-redundantly from NCBI and GTDB.

**Authors' Changes in Manuscript:** As mentioned in the authors' response.

---

## Author Comment (AC2)

**Authors' Response to Anonymous Referee # 2**

**General Comments**

**Referee's Comment:** In this manuscript, Bhattacharya et al., investigate and compare microbial community structure, metagenomes and metatranscriptomes in sediments below perennial and seasonal oxygen minimum zones in the Arabian Sea. They report the difference in community structure between the two as well as varying biogeochemical processes determined by -omics data and physiochemical and isotope data. Specifically, the perennial OMZ sediments exhibited cryptic methane cycle and high abundance of methanogens, ANME, SRB, and acetogens in surface sediments. Authors also incorporated physiochemical and geochemical data to help explain these processes and main environmental drivers of the community differences and I really appreciate this approach.

**Authors' Response:** We thank the Reviewer for appreciating the phenomenology and scientific approaches underlying the study.

**Authors' Changes in Manuscript:** Not applicable

Sampling and methodological approach were rigorous, well executed, and generated a plethora of thought-provoking and really valuable data. However, the manuscript is way too long and thus very difficult to get through. It also includes a lot of redundant information between the methods, results, and discussion sections (results in discussion/discussion in results). The main findings are difficult to follow due to the length of text and not very informative data presentation. Additionally, extensive lists of taxa make the manuscript really long and disturb the flow (without really bringing much valuable information). Seems like metatransciptome results (arguably one the most interesting parts of this study) are buried in text and not represented in figures. On the other hand, diversity data, which is valuable but not as captivating, is discussed at lengths and displayed in a large panel figure. I think the manuscript needs to be condensed and only the really interesting, major findings should be discussed and highlighted. I understand that sifting through so much data and determining what really matters is difficult, but I think it is a necessity so that this manuscript is comprehensible and thought provoking.
Figures should also be improved and better represent the compelling data the authors have here, but these data are buried in text.

**Authors' Response:** We agree with the concerns, so have now dealt with each of them in the revised manuscript.

**Authors' Changes in Manuscript:** Over the revision we have now
  - reduced the length of the manuscript considerably,
  - removed redundancy of information,
  - moved all collateral information such as the lists of taxa to the SI,
  - incorporated new figures for better visualization of metatranscriptomic and slurry incubation results,
  - deleted the detailed diversity data, keeping only what is central to demonstrate that the microbiome profiles of the two sediment systems are different (notably Figure 2 is central in this regard, because only this kind of an all-encompassing heat map can comprehensively and quantitatively illustrate that the microbiome profiles of the perennial and seasonal OMZ sediments are distinct, which in turn is one of the main discoveries of this paper),
  - removed the detailed values for the geochemical data,
  - and, overall, streamlined the results and discussions as per the objectives spelt out in Introduction.

**Specific Comments**

Line 53-58: please shorten this sentence or break it up, it is hard to follow

**Authors' Response:** We agree, now fixed.

**Authors' Changes in Manuscript:** The sentence has been simplified.

Line 67: maybe co-exist instead of co-founded?

**Authors' Response:** The idea here is to convey the sense that microbial communities get established (founded) in the sediments after getting deposited alongside organic matter. The word "exist" would not exactly convey this sense.

**Authors' Changes in Manuscript:** Not applicable.

Line 78-81: I am not sure this statement is true; I feel like there has been plenty of research done on this globally

**Authors' Response:** Flux and composition of the organic matter and microflora deposited has been shown to influence the carbon remineralization/sequestration dynamics of a sediment system (Kristensen et al., 1995; Parkes et al., 2000; Burdige, 2007; LaRowe et al., 2020). Water-column oxygenation level is also known to impact the preservation of labile (biochemically reactive) organic matter, as well as the composition of the microbial community in the seafloor (Jessen et al. 2017).

From this perspective, however, the following issues are almost unexplored,
- how microbiome profile changes in the age-depth context of a diagenetically maturing sediment package (Kallmeyer et al., 2012; Orsi et al., 2017),
- how differential bottom-water DO concentration, sedimentation rate, and flux and nature of the deposited organic matter, as often encountered along water-depth transects across continental margins (Middelburg, 2019a, 2019b), shape the microbiome profile of marine sediments.

**Authors' Changes in Manuscript:** The text has now been edited by incorporating new reference in such a way as to explain the above point more objectively.

Line 160: "DNA for metagenomic analysis was extracted"

**Authors' Response:** We agree and have now edited the text as suggested.

**Authors' Changes in Manuscript:** "Metagenomes were extracted" has been replaced with "DNA for metagenomic analysis was extracted".

Line: 225-250: This is so much text and it's hard to follow/keep track of. Maybe present in a table in SI? The manuscript is already so voluminous...

**Authors' Response:** We agree and have now moved these lists of taxa to the SI.

**Authors' Changes in Manuscript:** As mentioned in the authors' response.

Line 270-280: I don't think it is necessary to describe and discussion diversity indices at such length.

**Authors' Response:** We agree, so have now shifted the mathematical details to the SI and provided only the conceptual explanations in the main text.

**Authors' Changes in Manuscript:** As mentioned in the authors' response.

Line 312: have you checked GTDB for ANME?

**Authors' Response:** We have now upgraded all the metagenomic and metatranscriptomic data pertaining to ANME with analyses based on the searching of our sequence datasets against a comprehensive database of ANME genomes (MAGs) curated non-redundantly from NCBI and GTDB.

**Authors' Changes in Manuscript:** As mentioned in the authors' response.

Lines 497-508: At this point I am lost in the jargon and equations.

**Authors' Response:** We agree, so have conceptually elaborated and explained the mathematical terminologies by adding simpler descriptions.
In this context it is noteworthy that the probability density functions (equation fitting) determined and presented for the individual data series critically authenticate the phenomenon "synchronous population fluctuation of multiple metabolic types along the perennial OMZ cores", a main finding of this paper.
If we don't check the population fluctuation trends for potential equations and interpret them mathematically, people would not accept what we are calling increases, decreases or a bell-shaped (Gaussian) distributions are indeed so. From a statistical point of view, people would argue that if still finer resolution data are generated then what is the probability that the trends we are seeing now would still remain so. To answer this it is indispensable to check out probability density functions for the trends observed; if the equations were not obtainable even after carrying out the statistical tests then the envisioned phenomenon of synchronous population fluctuation for the different anaerobic metabolic types along the perennial OMZ sediment cores would have been completely on shaky ground.

**Authors' Changes in Manuscript:** We have now conceptually elaborated and explained the mathematical terminologies by adding simpler descriptions.

Line 612: The discussion specifically needs to be shortened and to the point. I think it will greatly improve the manuscript.

**Authors' Response:** We agree, and have now considerably streamlined the discussion section as per the objectives spelt out in the Introduction

**Authors' Changes in Manuscript:** As stated in the authors' response.

Fig 3: See my general comments above. Is this figure necessary in the main text?
Diversity is important but with the amount of data in this study I think transcriptome results or a map showing metabolic pathways would be much better to display here.

**Authors' Response:** We thank the reviewer for appreciating the significance of our metatranscriptomic data.
The following is a key phenomenological finding of this study. As a characteristic feature of the perennial-OMZ, overall microbial diversity peaks in the upper sediment-layer and then declines with increasing sediment-depth; microbiome profile is reverse in the seasonal-OMZ sediment horizon (this is stated in the Abstract also). Fig. 2 depicts this phenomenon in terms of the microbiome profile (community composition) along the three cores, while Fig. 3 presents the quantitative interpretation of microbial diversity along the three cores (i.e., in terms of Simpson and Shannon–Wiener indices calculated based on the phylum-level community composition data depicted in Fig. 2). Articulation of the aforesaid phenomenon without showing Figures 2 and 3 would make the inference sound baseless and subjective.

**Authors' Changes in Manuscript:** We have now incorporated new figure for the visualization of the results pertaining to the metatranscriptome analyses.

Fig 5: I am not sure this is the best way to represent your data

**Authors' Response:** We agree that this MATLAB-derived representation of the population fluctuation trends is atypical as the data is also unusual. However, it is the only way we can comprehensively compare population fluctuation patterns across the different metabolic categories as well as the distinct sediment cores.

**Authors' Changes in Manuscript:** Not applicable

Fig 6: Thank you for including geochemical data!

**Authors' Response:** We thank the Reviewer for appreciating the cross-disciplinary depth of our study.

**Authors' Changes in Manuscript:** Not applicable

---

## Author Response (AR1)

05 July 2021

Dear Professor Treude,

Thank you very much for allowing us to revise and re-submit our paper titled **Sedimentation rate and organic matter dynamics shape microbiomes across a continental margin** (bg-2021-25).

Over this revision we have dealt very positively with each of the comments that we had received, and in doing so,

- re-structured the text by deleting all mathematical jargons and simplifying the language,
- upgraded the data wherever it was warranted (for instance, in relation to the metagenome and metatranscriptome analyses for ANME-related reads),
- improved the presentation of the results by incorporating new figures for better visualization (of metatranscriptomic and slurry incubation data), and
- deleted, or moved to the SI, all the asides, or supportive details, thereby focusing the manuscript more towards the main findings of the study.

Please find our responses to all queries and suggestions in the Response to Reviewer files, and changes to the text shown in the Track Changes version of the manuscript.

We believe the Reviewers' comments have greatly improved the quality of the manuscript so as bring it at par with the high standards of *Biogeosciences*.

Yours, with regards
Wman
* * *
Dr. Wriddhiman Ghosh
Associate Professor
Department of Microbiology
Bose Institute
P-1/12 CIT Scheme VIIM
Kolkata 700054, India

**Authors' Response to Anonymous Referee #1**

**General comments**

**Referee's Comment:** Bhattacharya and colleagues obtained a very impressive set of metagenomic and metatranscriptomic data in combination with physicochemical and isotope geochemical measurements. The authors investigate differences in the sediment communities of permanent and seasonal oxygen minimum zones and present shifts in community structure and gene transcripts with depth as well as between the sites. The sampling, the generation, processing and analysis of the data, as well as the sequence archiving was performed carefully. Yet the presentation of the results needs to be improved. In my opinion, the manuscript needs to be shortened (substantially), better structured, and jargon needs to be avoided. The interesting and relevant messages of the study are buried in details, percentages and values. The study is difficult to access in part because the manuscript lacks visualizations (e.g. of the gene transcript abundance). It is recommended to structure all manuscript sections based on compelling objectives, find a red thread that guides the reader through these amazing data, and present only those results that are important to the major findings and conclusions and remove everything else. Please find detailed suggestions below.

**Authors' Response:** We thank the reviewer for appreciating the phenomenology and scientific approaches underlying the study. We also agree with the concerns of the reviewer, so have now dealt with each one of them in the revised manuscript.

**Authors' Changes in Manuscript:** In the Revised Manuscript we have
- re-structured the text by simplifying the language,
- improved the presentation of the results involving percentages and values (by incorporating new figures for better visualization of metatranscriptomic and slurry incubation data),
- deleted, or moved to the SI, all the asides, or supportive details, thereby focusing the manuscript more towards the main findings of the study.

**Specific comments**

**Referee's Comment:** A picture speaks a thousand words! That is also true for figures. Instead of listing all the taxa, concentrations, abundances etc, please try to visualize. It is so much easier to grasp the critical information that way. For example, the transcript data: The authors have such an incredible dataset. I am wondering why the impressive gene transcript abundance data is not shown in a dedicated figure. The trends become much clearer when all these abundances are shown along the depth profile, or across samples. For inspiration on representation of transcriptomic data check, e.g., these papers:
https://sfamjournals.onlinelibrary.wiley.com/doi/10.1111/1462-2920.14806
https://sfamjournals.onlinelibrary.wiley.com/doi/full/10.1111/1462-2920.13895
https://microbiomejournal.biomedcentral.com/articles/10.1186/s40168-019-0664-z

**Authors' Response:** We agree, and have now edited the manuscript as suggested.

**Authors' Changes in Manuscript:** In the Revised Manuscript we have now
- incorporated new figures for the visualization of the results pertaining to the slurry incubation experiments and metatranscriptome analyses (numbered as Figures 6 and 7 in the revised manuscript),
- most of the lists of taxa have now been deleted or shifted to the SI (please see lines 244-257, 266-268, 423-427 and 430-435 of the Track Changes file),
- numerical values for abundances, concentrations, etc. have been removed from the text as much as possible (for instances please see lines 417-419, 421-427, 428-435 and 797-825 of the Track Changes file)

**Referee's Comment:** L557ff: Can those methane concentrations be shown in a figure?

**Authors' Response:** We agree, now fixed.

**Authors' Changes in Manuscript:** The new figure number 6 graphically represents all the concentration values that were involved in the slurry incubation data.

**Referee's Comment:** L583ff: For me, and I think most other readers, it is very difficult to grasp all these percentages and make sense of them, compare them, interpret them. Can this be visualized as well?

**Authors' Response:** We agree, now fixed.

**Authors' Changes in Manuscript:** The new figure number 7 graphically represents all the percentage values that were involved in the metatranscriptome analyses.

**Referee's Comment:** Please try to avoid all terms that are very specific for a small area of science (jargon). L479-L489: This section is full of jargon: discontinuous reduction, Gaussian distribution, exponential-decay zone, probability density function, etc. Please avoid using these terms, because most of the readers will not know what they mean. Text sections become very difficult if not impossible to understand when they are full of jargon, and then the entire manuscript becomes hard to digest. In addition, when a reader gets lost in these details the overall message gets lost too. The rule of thumb here is: If for example the Gaussian distribution is really important for the study then it needs to be elaborated and explained with words, concepts and examples everybody understands. If it is not critical for the study, remove it, or move it to the SI for the experts that may be interested in the details.

**Referee's Comment:** L490-517: This section too is very hard to digest. Is all this detail really necessary? To be honest, I got lost in the equations right away, and was unable to say what the message of the paragraph is even after reading it repeatedly (I couldn't see the proverbial forest from all the trees). Please consider a broad audience.

L520-545: Same in this paragraph

**Authors' Response:** We agree with the common essence of the above three points, so have now removed all the equations and mathematical terminologies from the main text, which is now purely aimed at a broad audience of biogeochemistry.

This said, the mathematical definitions (fitted functions) authenticate the fluctuation trends recorded for the relative abundances of the different metabolic types, along the sediment cores explored.

If we don't check the population fluctuation trends for potential equations and interpret them mathematically, people would not accept what we are calling increases, decreases or bell-shaped patterns as indeed so. From a statistical point of view, people would argue that if still finer resolution data are generated then what is the probability that the trends we are seeing now would still remain so. To answer this it is crucial to check out mathematical functions for the trends observed.

**Authors' Changes in Manuscript:** Mathematical definitions (fitting to approximate functions) of all the fluctuation trends recorded for the relative abundances of the individual metabolic-types along the different sediment cores have now been given as Supplementary Note 3 in the SI file. In that SI note we have now conceptually elaborated the mathematical terminologies by adding simpler descriptions and examples. The components of all the equations have also been explained in the new SI note. The previous Figure 5 (and its legend containing the $\chi^2$ value for the individual functions fitted to the different fluctuation trends) is also a

part of the SI note. Please see lines 273-274, 277, 511-520, 532-538, 548-550, 524-526 563-573 and 1229-1252 of the Track Changes file.

**Referee's Comment:** In addition to an amazing dataset the authors have some very interesting findings. E.g. (L635:"coexistence and covariance of sulfate-reducers with methanogens, ANME and acetogens"). However, those too get lost in all the details. It is highly recommended to shorten and declutter the manuscript. Find out what are the most interesting findings, what is the red thread that keeps the reader's (and writers) attention. Then start by introducing compelling objectives, and then follow these objectives in the results and discussion. Everything that is not needed for the core story should be removed or moved to the SI. There is a lot of potential to shorten the manuscript, I would say it should be at least a third shorter, if not half.

**Authors' Response:** We agree, and have now considerably streamlined the discussion and shortened the manuscript.

**Authors' Changes in Manuscript:** As stated for all the general and specific comments.

**Referee's Comment:** The authors often include results in the discussion section. These sections should either be separated, i.e. the discussion only refers to the results, but contains very little values and numbers anymore. Or they should be combined in a Results and Discussion section. In any case, it would greatly help if the discussion was structured based on objectives that were introduced earlier. Presently the discussion is very broad, and I could not see where the readers attention/focus is drawn towards, i.e. what are the major take home messages.

**Authors' Response:** We agree, and have now streamlined the discussion as per the objectives of the work, the data obtained, and the phenomena envisioned. We have also made sure that there are no major results in this section anymore. Only a few data that were solely relevant in the background (perspective) of the discussions (phenomena envisioned) have been retained here in the form of Supplementary Tables or cited facts.

**Authors' Changes in Manuscript:** We have now removed the previous section "**4.2 Microbial community dynamics within the shallow SMTZ of SSK42/6**" from the Discussion to the Results as a new section 3.7 (please see lines 658-677 and 780-795 of the Track Changes file). We have also deleted all the geochemical data that were previously under "**4.3 Comparable pore-fluid chemistries of the AS_pOMZ and AS_sOMZ sediments**" and instead put in only a one-line summary for essence of the pore fluid chemistry of the sediment cores explored (please see lines 797-825 and 830-836 of the Track Changes file).

**Referee's Comment:** L615-630: This reads like a summary of the results section. In general, this is welcomed, as now the trends become clear, but it is still a results part, a discussion goes beyond that.

**Referee's Comment:** L700ff: This contains a lot of results again.

**Authors' Response:** We thank the reviewer for appreciating the importance of having this recapitulative paragraph to spell out the final trends clearly.

**Authors' Changes in Manuscript:** After this broad clarification of the microbiome architecture, we have streamlined the discussion in sync with the objectives of the work, the data obtained, and the phenomena envisioned. Accordingly, the entire content of the first Discussion section
**4.1 Peculiar population ecology of anaerobic microorganisms as a signature of pOMZ sediments**
has now been broken down (decluttered) under the following subsections.
    **4.1.1 Overview of the microbiome architecture**
    **4.1.2 Analogous microbiome architectures in other marine sediment systems**
    **4.1.3 Cryptic methane cycling in the upper-sediment-layers of AS_pOMZ**

**4.1.4 Mechanistic bases of methanogens:sulfate-reducers coexistence**
**4.1.5 Potential biogeochemical role of ANSOB**

Please see this restructuring over lines 682-776 of the Track Changes file.

L715-742: Now this section even contains isotope geochemical results that have not yet been mentioned before. There is no mention of a context, so this entire section needs to be moved to the results section. Also, there is way too much information, too many numbers, percentages, etc.

**Authors' Response:** The entire section (previously "**4.3 Comparable pore-fluid chemistries of the AS_pOMZ and AS_sOMZ sediments**") has now been deleted from the revised manuscript. Since these are all published data (adequately cited as Fernandes et al., 2018, 2020) they cannot be put under Results; yet, their recapitulation is central to the present revelation of distinct microbiome structures and functions, amidst comparable pore fluid chemistries, in the sediment horizons underlying the seasonal (shallow coastal) and perennial (deep sea) oxygen minimum zones.

**Authors' Changes in Manuscript:** In the revised text, we have now removed all these geochemical data (please see lines 797-825 of the Track Changes file) and instead put in only a one-sentence summary (lines 830-836 of the Track Changes file) for the overall comparison of the pore fluid chemistry of the sediment cores explored (the relevant figure is still there to serve all quantitative purposes).

L757ff: Results

**Authors' Response:** These few lines are central to the main postulation of this paper: **Sedimentation rate and organic matter dynamics are the main drivers of microbiome architecture.** However, since these are all published data (Fernandes et al., 2018, 2020, cited in the legend of the current Figure 5) they cannot be put under Results; we therefore have no other option but to recall these facts here only.

**Authors' Changes in Manuscript:** Not applicable.

**Technical Corrections**

**Referee's Comment:** L38: the influence of bottom water oxygen on benthic microbial communities is not entirely unknown (e.g. Jessen et al. Sci Adv 2017), so it is recommended to write "are less understood" or "are still unclear" etc.

**Authors' Response:** We agree, now fixed.

**Authors' Changes in Manuscript:** As mentioned in the authors' response (please see line 38 of the Track Changes file).

**Referee's Comment:** L81: This section should include mention of the study by Jessen et al 2017 Sci Adv, as it is presenting findings on microbial communities and organic matter in sediments along an oxygen gradient in the Black Sea

**Authors' Response:** We agree, Jessen et al. 2017 now mentioned.

**Authors' Changes in Manuscript:** As mentioned in the authors' response (please see lines 75-77 of the Track Changes file).

**Referee's Comment:** L105: what is meant by hectic? Do you mean substantial?

**Authors' Response:** "Hectic chemoorganoheterotrophic activities" now reworded as "high levels of chemoorganoheterotrophic activities".

**Authors' Changes in Manuscript:** As mentioned in the authors' response (please see lines 108-109 of the Track Changes file).

**Referee's Comment:** L119: ANME only refers to the ANaerobic MEthane oxidizing archaea.

**Authors' Response:** We agree, now fixed.

**Authors' Changes in Manuscript:** "bacteria symbionts" deleted (please see line 122 of the Track Changes file).

**Referee's Comment:** L167: Thank you for depositing the sequence data prior to submission, and also thanks for including all raw data and not just MAGs! That is best practice!

**Authors' Response:** We thank the reviewer for this positive reception.

**Authors' Changes in Manuscript:** Not applicable.

**Referee's Comment:** L227ff: These lists are a good place to start shortening the MS. It could be included in a supplementary table. The idea of summing up the taxa of a target guild may be clear to the reader without the need to present an exhaustive list.

**Authors' Response:** We agree, so have now moved these lists to the SI.

**Authors' Changes in Manuscript:** As mentioned in the authors' response (please see lines 244-257 and 266-268 of the Track Changes file).

**Referee's Comment:** L251: What about ANME-2d (Methanoperedens). It becomes increasingly apparent that this clade may be important at marine systems as well. E.g. in this very recent publication https://www.nature.com/articles/s41396-021-00918-w

**Authors' Response:** We have now upgraded all the metagenomic and metatranscriptomic data pertaining to ANME with analyses based on the searching of our sequence datasets against a comprehensive database of ANME genomes (MAGs) curated non-redundantly from NCBI using the accession numbers given in GTDB (please see lines 187-195, 538-547, 551-560, 632-638 and 725-728 of the Track Changes file, plus the modified Figure 4 and new Table S4). The new Table S4 shows the composition of the revised ANME database.

**Authors' Changes in Manuscript:** As mentioned in the authors' response.

**Referee's Comment:** L270: The equations for the reported Simpson and Shannon indices seem to be the standard equation used for these respective indices. Is there a reason why the authors chose to present the math in detail? Here too it may suffice to cite the respective papers and maybe elaborate on these metrics in the SI. For the reader it would be more helpful to explain the meaning of these metrics, their end members (min/max values) and how to interpret them. In my opinion it is more necessary to explain the terms, .e.g. evenness/equitability than show the equation. As an example, for me it helps to interpret a given index, if I am aware of the max/min values. E.g. that the minimum value Inverse
Simpson can be equals 1 (a pure culture) and the maximum value equals the number of observed species (which is the case when all species occur at exactly the same relative proportion).

**Authors' Response:** We agree, so have now removed all the commonplace mathematical details (including the equations) from the manuscript.

**Authors' Changes in Manuscript:** As mentioned in the authors' response (please see lines 291-304 of the Track Changes file).

**Referee's Comment:** L312: From which studies did the two ANME genomes come from? Please reference. There should be quite a number of available ANME MAGs by now. Maybe these can be used as a database for your next studies.

**Authors' Response:** As mentioned above, we have now upgraded all the metagenomic and metatranscriptomic data pertaining to ANME with analyses based on the searching of our sequence datasets against a comprehensive database of ANME genomes (MAGs) curated non-redundantly from NCBI using the accession numbers given in GTDB. Accordingly, the web address (https://gtdb.ecogenomic.org/) and reference for GTDB (Parks et al., 2018) has been cited in the revised manuscript. Furthermore the new Table S4, which shows the composition of the genomic sequence database curated for ANME, cites the GenBank accession numbers of the genome / fosmid sequences included.

**Authors' Changes in Manuscript:** As mentioned in the authors' response (please see lines 187-195, 538-547, 551-560, 632-638 and 725-728 of the Track Changes file, plus the modified Figure 4 and new Table S4)

**Referee's Comment:** L328: Was hydrogen added as an electron donor for hydrogenotrophic methanogens?

**Authors' Response:** Formate present in the medium (4 g L$^{-1}$ HCOONa) acts as the main hydrogen source (please see line 354 of the Track Changes file). Additionally, the headspaces of the culture bottles contained hydrogen (10% v/v) as a mixture of $N_2$:$CO_2$:$H_2$ = 80:10:10 (v/v/v) was used to deoxygenate the medium, as well as to preset the Whitley H35 Hypoxystation (inside which inoculum addition, culture incubation and culture transfer were carried out) at 0% partial pressure of $O_2$.

**Authors' Changes in Manuscript:** The second point was already there in the reference that was cited for the methanogen-specific medium used (Whitman et al., 2006); still we have now mentioned this in the revised text over and above the existing reference (please see lines 358-360 of the Track Changes file).

**Referee's Comment:** L356ff: A lot of the results in this section are redundant to the Sampling section in the Materials and Methods. Please shorten, or remove, at either section.

**Authors' Response:** We agree, now fixed.

**Authors' Changes in Manuscript:** Redundant information now removed from the Results section (please see lines 384-390 of the Track Changes file).

**Referee's Comment:** L364: What does BP stand for, please write out each abbreviation at first use.

**Authors' Response:** We agree, now fixed.

**Authors' Changes in Manuscript:** Full form of BP (before present) has now been provided (please see line 392 of the Track Changes file).

**Referee's Comment:** L379: Which database and version was used for the taxonomic classification? I might have missed this in the methods, but if it is not mentioned, please include.

**Authors' Response:** For taxonomic classification of metagenomic reads the datasets were searched separately against the NCBI non-redundant (*nr*) protein sequence database (last access: 14 April 2020) as well as the four distinct databases of single-copy conserved marker proteins, which were specially curated from CheckM version 0.7.060 (Parks et al., 2015, last access: 22 December 2020).

**Authors' Changes in Manuscript:** This was already mentioned in Methods (please see the unchanged lines 179-187 of the Track Changes file).

**Referee's Comment:** L387: Please refer to "relative sequence abundance" to avoid confusion with other measures of relative abundance.

**Authors' Response:** We agree, now fixed.

**Authors' Changes in Manuscript:** Text edited as suggested (please see the unchanged line 408 of the Track Changes file).

**Referee's Comment:** L392f: Generally, it is recommended to phrase it the other way around: "…xx % of metagenomic reads affiliated with Alphaproteobacteria…"

**Authors' Response:** This, as well as several other similar lines containing taxon names and percentage values for relative sequence abundance of taxa, has now been deleted from the revised text (please see lines 417-419, 421-427 and 428-435 of the Track Changes file).

**Authors' Changes in Manuscript:** As mentioned in the authors' response.

**Referee's Comment:** L393ff: Personally, I think it is not necessary to list all these taxa. It makes the manuscript very long and voluminous and disrupts the reading flow. Unless a specific taxon needs to be highlighted, it will suffice to point the reader to a figure showing the clades and their relative sequence abundances.

**Authors' Response:** We agree and have now reduced the text as suggested.

**Authors' Changes in Manuscript:** Mentions of all the relatively-less abundant taxa have been removed from the revised text (please see the unchanged lines 423-427 and 430-435 of the Track Changes file) and the overall differences in the microbiome profiles of perennial and seasonal OMZ sediments have been pointed out by referring to the heat map of Figure 2.

**Referee's Comment:** L480: I think microbial guilds would be a more widespread and better term than "metabolic-types"

**Authors' Response:** We agree that microbial guilds is a widespread terminology but since it does not specify that the groups here were unified based on particular metabolisms we prefer to retain the term "metabolic-types".

**Authors' Changes in Manuscript:** Not applicable.

**Referee's Comment:** L579: The number of reads is important, but not here. 1. Large numbers break the reading flow and 2.) including the read numbers does not add info that is necessary for the argument at this point. This section is about mcrA transcripts and so it should be focused on this.

**Authors' Response:** We agree, and have now moved the information on read numbers to Methods section.

**Authors' Changes in Manuscript:** As mentioned in the authors' response. Please see lines 317-319 and 624-627 of the Track Changes file.

**Referee's Comment:** L580ff: redundant to materials section. Please remove.

**Authors' Response:** We have already moved the information on read numbers to the Methods section, but we believe that a sentence such as the one given here is essential to connect the subsequent data flow to the procedure adopted - "Overall, the two metatranscriptomic datasets were mapped separately against the comprehensive genomic sequence databases curated individually for not only the methanogens but also the ANME, sulfate-reducers, acetogens and ANSOB".

**Authors' Changes in Manuscript:** Not applicable.

**Referee's Comment:** L652: Could the absence of methane not be explained also by aerobic methane oxidation?
Did you check if you have aerobic methane oxidizers in the dataset (eg Methylococcales).

**Authors' Response:** We agree and thank the reviewer for vindicating our previous findings regarding aerobic methane oxidation in the sediment cores SSK42/5 and SSK42/6 (Bhattacharya et al.: FEMS Microbiol. Letts., 367, fnaa157, 2020). We have also checked the assembled and annotated metatranscriptomic data for aerobic methane oxidation genes.

**Authors' Changes in Manuscript:** The revised text now factors in the role of aerobic methane oxidizers. Please see lines 728-731 of the Track Changes file.

**Referee's Comment:** L659: Is there a reason why a database like GTDB (https://www.nature.com/articles/nbt.4229?testing) was not used? It should contain plenty of ANME genomes/MAGs.

**Authors' Response:** We have now upgraded all the metagenomic and metatranscriptomic data pertaining to ANME with analyses based on the searching of our sequence datasets against a comprehensive database of ANME genomes (MAGs) curated non-redundantly from NCBI using the accession numbers given in GTDB. The new Table S4 shows the composition of the revised ANME database.

**Authors' Changes in Manuscript:** As mentioned in the authors' response (please see lines 187-195, 538-547, 551-560, 632-638 and 725-728 of the Track Changes file, plus the modified Figure 4 and new Table S4).

**Authors' Response to Anonymous Referee # 2**

**General Comments**

**Referee's Comment:** In this manuscript, Bhattacharya et al., investigate and compare microbial community structure, metagenomes and metatranscriptomes in sediments below perennial and seasonal oxygen minimum zones in the Arabian Sea. They report the difference in community structure between the two as well as varying biogeochemical processes determined by -omics data and physiochemical and isotope data. Specifically, the perennial OMZ sediments exhibited cryptic methane cycle and high abundance of methanogens, ANME, SRB, and acetogens in surface sediments. Authors also incorporated physiochemical and geochemical data to help explain these processes and main environmental drivers of the community differences and I really appreciate this approach.

**Authors' Response:** We thank the Reviewer for appreciating the phenomenology and scientific approaches underlying the study.

**Authors' Changes in Manuscript:** Not applicable

Sampling and methodological approach were rigorous, well executed, and generated a plethora of thought-provoking and really valuable data. However, the manuscript is way too long and thus very difficult to get through. It also includes a lot of redundant information between the methods, results, and discussion sections (results in discussion/discussion in results). The main findings are difficult to follow due to the length of text and not very informative data presentation. Additionally, extensive lists of taxa make the manuscript really long and disturb the flow (without really bringing much valuable information). Seems like metatransciptome results (arguably one the most interesting parts of this study) are buried in text and not represented in figures. On the other hand, diversity data, which is valuable but not as captivating, is discussed at lengths and displayed in a large panel figure. I think the manuscript needs to be condensed and only the really interesting, major findings should be discussed and highlighted. I understand that sifting through so much data and determining what really matters is difficult, but I think it is a necessity so that this manuscript is comprehensible and thought provoking.
Figures should also be improved and better represent the compelling data the authors have here, but these data are buried in text.

**Authors' Response:** We agree with the concerns, so have now dealt with each of them in the revised manuscript.

**Authors' Changes in Manuscript:** Over the revision we have now
- reduced the length of the manuscript considerably,
- removed redundancy of information,
- moved all collateral information such as the lists of taxa to the SI,
- incorporated new figures for better visualization of metatranscriptomic and slurry incubation results,
- deleted the detailed diversity data, keeping only what is central to demonstrate that the microbiome profiles of the two sediment systems are different (notably Figure 2 is central in this regard, because only this kind of an all-encompassing heat map can comprehensively and quantitatively illustrate that the microbiome profiles of the perennial and seasonal OMZ sediments are distinct, which in turn is one of the main discoveries of this paper),
- removed all the numerical values for the geochemical data and put in only a one-line summary for the pore fluid chemistry of the sediment cores explored,
- and, overall, streamlined the results and discussions as per the objectives spelt out in Introduction.

<h1 align="center">Specific Comments</h1>

**Line 53-58: please shorten this sentence or break it up, it is hard to follow**

**Authors' Response:** We agree, now fixed.

**Authors' Changes in Manuscript:** The sentence has been restructured. Please see lines 53-59 of the Track Changes file.

**Line 67: maybe co-exist instead of co-founded?**

**Authors' Response:** The idea here is to convey the sense that microbial communities get established (founded) in the sediments after getting deposited alongside organic matter.

**Authors' Changes in Manuscript:** "Co-founded" has now been replaced with "co-established". Please see line 67 of the Track Changes file.

**Line 78-81: I am not sure this statement is true; I feel like there has been plenty of research done on this globally**

**Authors' Response:** Flux and composition of the organic matter and microflora deposited has been shown to influence the carbon remineralization/sequestration dynamics of a sediment system (Kristensen et al., 1995; Parkes et al., 2000; Burdige, 2007; LaRowe et al., 2020). Water-column oxygenation level is also known to impact the preservation of labile (biochemically reactive) organic matter, as well as the composition of the microbial community in the seafloor (Jessen et al. 2017).

From this perspective, however, we do not have any direct and comprehensive idea about the following:

- how microbiome profile changes in the age-depth context of a diagenetically maturing sediment package (some idea in this regard can be obtained from Kallmeyer et al., 2012; Orsi et al., 2017, but that does not encompass comprehensive microbiome profiles),
- how differential bottom-water DO concentration, sedimentation rate, and flux and nature of the deposited organic matter, as often encountered along water-depth transects across continental margins (Middelburg, 2019a, 2019b), shape the microbiome profile of marine sediments.

**Authors' Changes in Manuscript:** The text has now been edited (by incorporating new reference) in such a way as to explain the above point more objectively. Please see lines 73-85 of the Track Changes file.

**Line 160: "DNA for metagenomic analysis was extracted"**

**Authors' Response:** We agree and have now edited the text as suggested.

**Authors' Changes in Manuscript:** "Metagenomes were extracted" has been replaced with "DNA for metagenomic analysis was extracted". Please see line 162 of the Track Changes file.

**Line: 225-250: This is so much text and it's hard to follow/keep track of. Maybe present in a table in SI? The manuscript is already so voluminous...**

**Authors' Response:** We agree and have now deleted, or moved to the SI, all the asides, or supportive details, thereby focusing the manuscript more towards the main findings of the study.

**Authors' Changes in Manuscript:** As mentioned in the authors' response (please see lines 226-232, 244-257 and 266-268 of the Track Changes file).

Line 270-280: I don't think it is necessary to describe and discussion diversity indices at such length.

**Authors' Response:** We agree, so have now removed all the commonplace mathematical details from the manuscript.

**Authors' Changes in Manuscript:** As mentioned in the authors' response (please see lines 291-304 of the Track Changes file).

Line 312: have you checked GTDB for ANME?

**Authors' Response:** We have now upgraded all the metagenomic and metatranscriptomic data pertaining to ANME with analyses based on the searching of our sequence datasets against a comprehensive database of ANME genomes (MAGs) curated non-redundantly from NCBI using the accession numbers given in GTDB. The new Table S4 shows the composition of the revised ANME database.

**Authors' Changes in Manuscript:** As mentioned in the authors' response (please see lines 187-195, 538-547, 551-560, 632-638 and 725-728 of the Track Changes file, plus the modified Figure 4 and new Table S4).

Lines 497-508: At this point I am lost in the jargon and equations.

**Authors' Response:** We agree, so have now removed all the equations and mathematical terminologies from the main text, which is now purely aimed at a broad audience of biogeochemistry.

This said, the mathematical definitions (fitted functions) authenticate the fluctuation trends recorded for the relative abundances of the different metabolic types, along the sediment cores explored.

If we don't check the population fluctuation trends for potential equations and interpret them mathematically, people would not accept what we are calling increases, decreases or bell-shaped patterns as indeed so. From a statistical point of view, people would argue that if still finer resolution data are generated then what is the probability that the trends we are seeing now would still remain so. To answer this it is crucial to check out mathematical functions for the trends observed.

**Authors' Changes in Manuscript:** Mathematical definitions (fitting to approximate functions) of all the fluctuation trends recorded for the relative abundances of the individual metabolic-types along the different sediment cores have now been given as Supplementary Note 3 in the SI file. In that SI note we have now conceptually elaborated the mathematical terminologies by adding simpler descriptions and examples. The components of all the equations have also been explained in the new SI note. The previous Figure 5 (and its legend containing the $\chi^2$ value for the individual functions fitted to the different fluctuation trends) is also a part of the SI note. Please see lines 273-274, 277, 511-520, 532-538, 548-550, 524-526 563-573 and 1229-1252 of the Track Changes file.

Line 612: The discussion specifically needs to be shortened and to the point. I think it will greatly improve the manuscript.

**Authors' Response:** We agree, and have now streamlined the discussion section as per the objectives of the work spelt out in the Introduction, the data obtained, and the phenomena envisioned.

**Authors' Changes in Manuscript:** We have now removed the previous section "**4.2 Microbial community dynamics within the shallow SMTZ of SSK42/6**" from the Discussion to the Results as a new section 3.7 (please see lines 658-677 and 780-795 of the Track Changes file). We have also deleted all the geochemical data that were previously under "**4.3 Comparable pore-fluid chemistries of the AS_pOMZ**

**and AS_sOMZ sediments**" and instead put in only a one-line summary for essence of the pore fluid chemistry of the sediment cores explored (please see lines 797-825 and 830-836 of the Track Changes file).

Furthermore, the first Discussion section
**4.1 Peculiar population ecology of anaerobic microorganisms as a signature of pOMZ sediments**
has now been broken down (decluttered) under the following subsections.
    **4.1.1 Overview of the microbiome architecture**
    **4.1.2 Analogous microbiome architectures in other marine sediment systems**
    **4.1.3 Cryptic methane cycling in the upper-sediment-layers of AS_pOMZ**
    **4.1.4 Mechanistic bases of methanogens:sulfate-reducers coexistence**
    **4.1.5 Potential biogeochemical role of ANSOB**

Please see this restructuring over lines 682-776 of the Track Changes file.

Fig 3: See my general comments above. Is this figure necessary in the main text?
Diversity is important but with the amount of data in this study I think transcriptome results or a map showing metabolic pathways would be much better to display here.

**Authors' Response:** We thank the reviewer for appreciating the significance of the diversity data as well as the metatranscriptome analyses.

The following is a key phenomenological finding of this study. As a characteristic feature of the perennial-OMZ, overall microbial diversity peaks in the upper sediment-layer and then declines with increasing sediment-depth; microbiome profile is reverse in the seasonal-OMZ sediment horizon (this is stated in the Abstract also). Fig. 2 depicts this phenomenon in terms of the microbiome profile (community composition) along the three cores, while Fig. 3 presents the quantitative interpretation of microbial diversity along the three cores (i.e., in terms of Simpson and Shannon–Wiener indices calculated based on the phylum-level community composition data depicted in Fig. 2). Articulation of the aforesaid phenomenon without showing Figures 2 and 3 would make the inference sound baseless and subjective.

**Authors' Changes in Manuscript:** We have now incorporated a new figure (numbered as 7 in the revised manuscript) for the visualization of the results pertaining to the metatranscriptome analyses. In the revised Methods and Results, we have now cut back considerably on the diversity details (please see lines 417-419, 421-427 and 428-435 of the Track Changes file) and also deleted the description of the indices as well as the relevant equations (please see lines 291-304 of the Track Changes file).

Fig 5: I am not sure this is the best way to represent your data

**Authors' Response:** We agree that this MATLAB-derived representation of the population fluctuation trends is atypical as the data is also unusual. However, it is the only way we can comprehensively compare the population fluctuation patterns (zones) across the different metabolic categories as well as the distinct sediment cores.

**Authors' Changes in Manuscript:** The previous Figure 5 (and its legend containing the $\chi^2$ value for the individual functions fitted to the different fluctuation trends) is no more in the main manuscript; it is now only a part of Supplementary Note 3 in the SI file.

Fig 6: Thank you for including geochemical data!

**Authors' Response:** We thank the Reviewer for appreciating the cross-disciplinary depth of the study.

**Authors' Changes in Manuscript:** As per another suggestion of Reviewer 1, in the revised text, we have now removed all these geochemical data (please see lines 797-825 of the Track Changes file) and instead

put in only a one-sentence summary (lines 830-836 of the Track Changes file) for the overall comparison of the pore fluid chemistry of the sediment cores explored (the relevant figure is still there to serve all quantitative purposes).